# Culturally Grounded Real-World Evaluation of Korean Vision–Language Models

## Abstract

VLMs perform well on standard benchmarks, yet their performance on authentic, culturally grounded tasks remains underexplored. We introduce HAERAE-VISION, a Korean real-world benchmark built from 86,052 question–image pairs across nine online platforms. Through a six-stage pipeline that applies appropriateness filtering, difficulty calibration, image dependency verification, checklist-based decomposition, and multi-phase human validation, we curate 653 rigorously validated items across 13 domains (0.76% survival). Each item is paired with a structured checklist rubric, enabling fine-grained evaluation beyond single-point correctness. We evaluate 39 VLMs spanning proprietary, open-weight, and Korean-specialized families under a unified protocol, and scoring with LLM judges demonstrates high reliability (Krippendorff's $\alpha = 0.867$). Even the strongest systems (Gemini 2.5 Pro, GPT-5) remain below 50% accuracy, with errors concentrated in explicitness and procedural reasoning, while Korean-specialized models show no clear advantage over multilingual counterparts. These findings highlight persistent gaps in real-world multimodal reasoning. Our work further offers a reproducible methodology for constructing robust, culturally grounded benchmarks across languages.

## 1 Introduction

Why another cultural benchmark? Cultural benchmarks have been studied in both language-only and multimodal settings (Kiela et al., 2021; Liang et al., 2022; Kim & Jung, 2025; Ju et al., 2024; Son et al., 2023). Much of this work, however, adopts a narrow view of culture that emphasizes shallow, factoid-style tasks (e.g., identifying foods such as kimchi) (Park et al., 2024; Jeong et al., 2025). We instead treat culture as a broader communicative context: how native speakers actually converse, including colloquialisms, slang, elliptical phrasing, and other pragmatic cues, not merely region-specific facts. Moreover, existing evaluations often present clean, fully contextualized questions, whereas real user queries are messy, under-specified, and informal. This mismatch likely explains part of the gap between benchmark scores and real-world VLM performance (Li et al., 2025).

Online communities offer a promising source of authentic data: they contain organically occurring multimodal questions that reflect users' real information needs and communicative styles (Chen et al., 2024). Korean platforms span cultural practices, technical forums, and everyday problem-solving, making them an ideal setting for a culturally grounded benchmark. In this work, we curate 86,052 question–image pairs and process them through a six-stage pipeline to construct **HAERAE-Vision**. Each instance is open-ended and paired with solution-specific checklists, and model outputs are evaluated with LLM-based judges (Kim et al., 2024; Lee et al., 2024; Bai et al., 2024). This design both builds on and departs from prior Korean benchmarks that are primarily multiple-choice (Hwang et al., 2025).

The resulting benchmark presents a substantially more challenging task than prior Korean datasets. Even the strongest proprietary models, Gemini 2.5 Pro and GPT-5, reach only 48.54 and 48.01, respectively. By contrast, earlier Korean benchmarks report notably higher scores with older-generation models such as GPT-4o (e.g., KRETA (Hwang et al., 2025): 84.6; K-VISCUIT (Park et al., 2024): 89.5; K-MMB: 81.01; K-SEED: 76.98; K-DTCBench: 85.80 (Ju et al., 2024)). These gaps suggest that **HAERAE-Vision** captures underrepresented phenomena and sets a more realis-

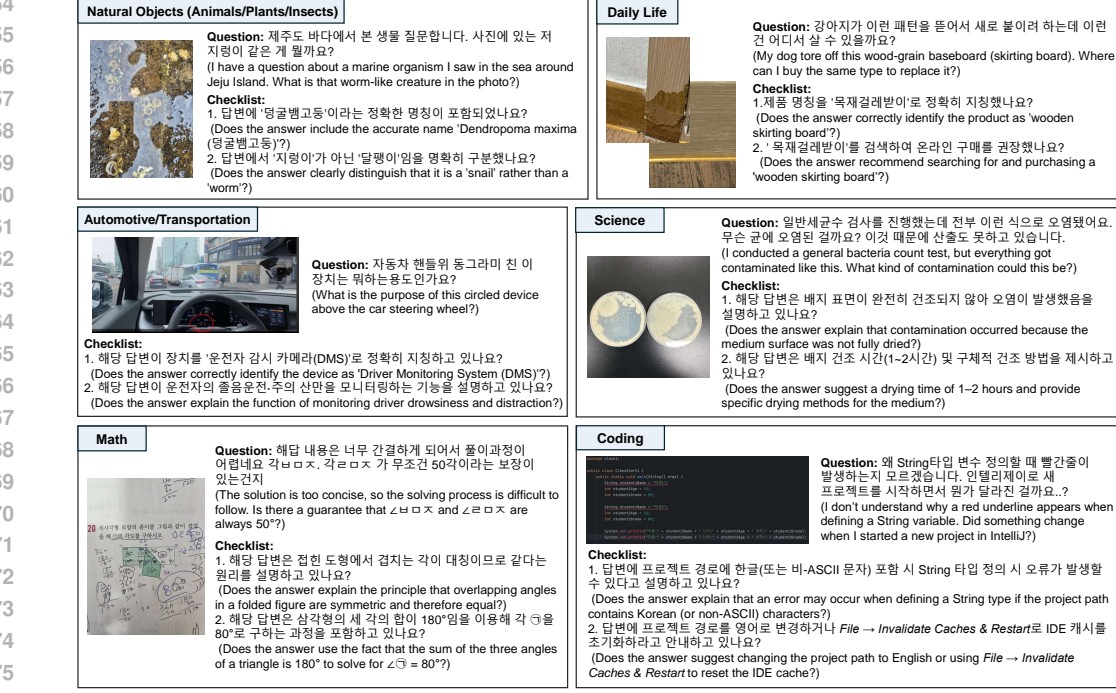

Figure 1: Representative examples from **HAERAE-Vision** across four of the 13 domains. Each example shows a Korean question with English translation, the corresponding image, and a truncated checklist of evaluation criteria (showing two items for brevity).

tic, demanding target for real-world performance that stresses not only cultural grounding but also general multimodal reasoning.

Our contributions are:

- **Authentic, curated data:** Starting from 86,052 question–image pairs across nine platforms, we apply rigorous filtering to produce 653 high-quality items covering 13 domains (0.76% survival).
- **Systematic quality pipeline:** A six-stage process combining appropriateness filtering, difficulty calibration, image dependency verification, checklist-based decomposition, and multi-phase human review.
- **Cultural grounding:** Tasks require Korean-specific knowledge (e.g., transport systems, cultural artifacts, colloquial visual cues) rarely represented in global benchmarks.
- **Replicable methodology:** A general pipeline and checklist-based evaluation framework that can be applied across languages and domains.

## 2 HAERAE-VISION

We present **HAERAE-Vision** not only as a benchmark but also as a methodological contribution: a principled, reproducible pipeline for transforming large-scale, noisy community data into high-quality multimodal evaluation problems. Our pipeline is intentionally designed to be generalizable across languages and cultures, enabling its direct application to other populations and domains.

### 2.1 DATASET CONSTRUCTION PIPELINE

We design a six-stage filtering pipeline that progressively removes noise while preserving authenticity and difficulty. Starting from 86,052 raw question–image pairs from nine Korean platforms, including general Q&A (KnowledgeIn), specialized communities (BRIC, Ruliweb, MonsterZym, Quasarzone), business platforms (i-Boss), and coding forums (Inflearn, Codeit, Okky), we obtain

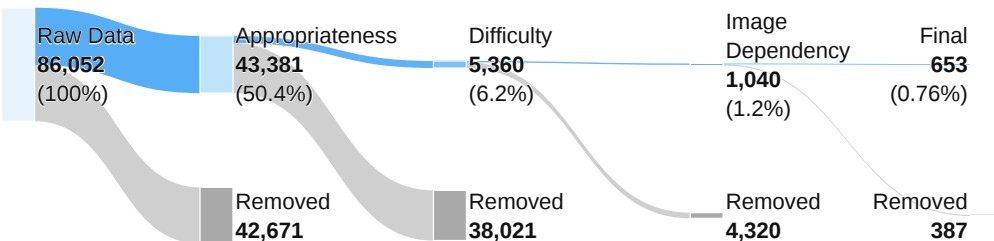

Figure 2: Count-based filtering funnel. Blocks show remaining items after each filter; percentages are relative to the raw set (86,052). The checklist-derivation step does not change counts and is omitted. The last block (*Human Validation*) reports the final dataset size (653).

653 high-quality evaluation problems (0.76% survival rate). Figure 2 illustrates the data reduction across stages.

**Stage 1: Data Collection.** Collect question–answer pairs containing images, prioritizing questions with adopted answers on KnowledgeIn (indicating asker-validated quality) and those with high engagement metrics (views, likes, comments) to capture questions that the community finds valuable.

**Stage 2: Appropriateness Assessment.** Each pair is screened along three axes: content safety (political/religious material, hate/discrimination, self-harm, adult content), objectivity (subjective or unverifiable prompts), and temporal stability (time-sensitive queries). Three tailored GPT-4o prompts return structured JSON flags for each axis. This stage filters out 42.7% of the raw data, ensuring that the remaining items are safe, verifiable, and temporally stable (Appendix B).

**Stage 3: Difficulty Calibration.** To prevent benchmark saturation, we remove questions that are trivially easy across models. Multiple strong models (GPT-4o, Gemini-1.5-Flash, Claude-3.5) are prompted with the ground-truth answer bundle, and their responses are scored for overlap (0–1). Items with consensus scores above 0.6 are excluded. This stage produces the largest reduction (87.6%), ensuring that the final dataset remains challenging even for state-of-the-art systems.

**Stage 4: Image Dependency Verification.** Verify that each question genuinely requires visual reasoning. For each item, a lightweight multimodal model (Gemini 2.0 Flash) generates answers with and without image access. An LLM rubric then labels the item as *image-required*, *no-image-needed*, or *uncertain* and assigns a 1–5 quality-gap score. Only items labeled as *image-required* are retained, ensuring that solving the problem cannot be reduced to text-only reasoning.

**Stage 5: Checklist Generation.** Convert answers into structured checklists containing 1–5 criteria, generated by o4-mini. These checklists capture the minimum necessary elements rather than exhaustive content, focusing on correctness, explanation, and reasoning steps. This design enables partial credit scoring and supports reproducible, automated evaluation across models.

**Stage 6: Human Validation.** Conduct a three-phase human validation with seven Korean-speaking annotators possessing relevant domain expertise. Phase 1 applies conservative filtering, removing any item flagged by either of two annotators. Phase 2 refines questions and regenerates or edits checklists for clarity. Phase 3 performs a final audit, consolidating categories and ensuring consistency. This rigorous process removes 31.4% of the remaining items, yielding a high-quality final set of 653 problems.

Together, these stages form a generalizable benchmark construction recipe that balances authenticity, difficulty, and cultural relevance. Beyond the Korean case, this pipeline can be directly applied to other languages and domains to produce culturally grounded multimodal benchmarks.

## 2.2 DATASET STATISTICS

Our final benchmark contains 653 problems with an average of 3.3 checklist items and 1.3 images per question, illustrating the multimodal nature of authentic community queries. Figure 3 presents the distribution across 13 categories, where Natural Objects and Gaming are the most represented, underscoring the visually oriented nature of community-driven questions. Platform survival rates

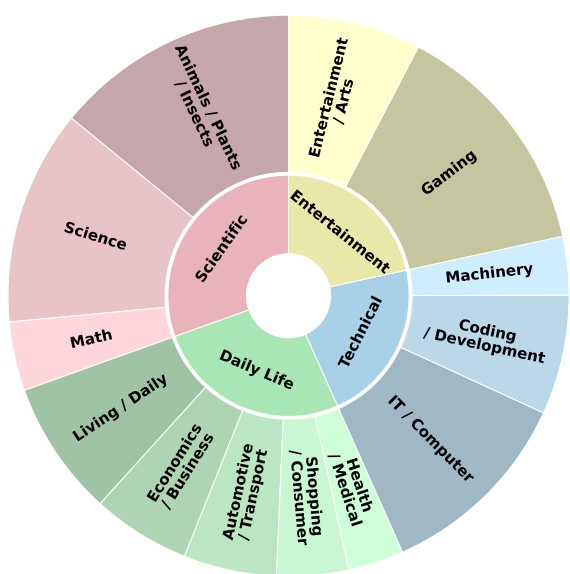

**Dataset statistics ($n=653$ questions)**

| Metric | Mean | Range |
|---|---|---|
| Q length (char) | 94.4 | 6–2,030 |
| Images per Q | 1.3 | 1–6 |
| Checklist items | 3.3 | 1–5 |

| Category | # Items | % |
|---|---|---|
| Gaming | 91 | 13.9 |
| Entertainment/Arts | 50 | 7.7 |
| Natural Objects | 92 | 14.1 |
| Science | 81 | 12.4 |
| Mathematics | 26 | 4.0 |
| IT/Computer | 75 | 11.5 |
| Coding/Development | 45 | 6.9 |
| Machinery | 22 | 3.4 |
| Daily Life | 51 | 7.8 |
| Business/Economics | 37 | 5.7 |
| Transportation | 35 | 5.4 |
| Shopping/Consumer | 27 | 4.1 |
| Health/Medical | 21 | 3.2 |
| **Total** | **653** | **100.0** |

Figure 3: Overview of **HAERAE-Vision**. Left: domain distribution across 13 categories. Right: summary statistics of question length, number of images, and checklist items, highlighting the diversity and multimodal nature of the benchmark.

vary significantly (0.2% to 14.4%), showing distinct community characteristics. Scientific communities show high content appropriateness but lower image dependency, while visual-oriented platforms like gaming communities demonstrate the opposite pattern (detailed breakdown in Appendix D).

## 2.3 KOREAN CULTURAL GROUNDING

Our benchmark includes questions that require distinctively Korean cultural and linguistic knowledge, setting it apart from translated or synthetic datasets. Figure 4 shows representative examples spanning traditional culture (Korean paintings with classical calligraphy), modern technology (Seoul subway navigation, TV interfaces), entertainment (recognizing Korean actors in drama scenes), and historical documents (family registries). These culturally embedded questions constitute about one quarter of the dataset (23.7%) and often demand knowledge beyond what is captured in general-purpose training corpora.

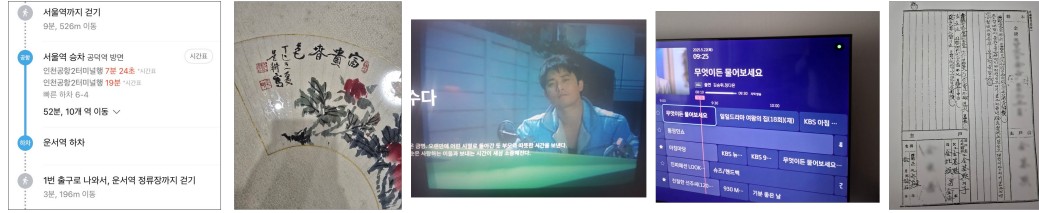

Figure 4: Examples highlighting the cultural specificity of **HAERAE-Vision**: (a) Seoul subway interface, (b) traditional painting with calligraphy, (c) Korean drama scene requiring celebrity recognition, (d) TV channel settings, (e) historical family registry. Such culturally grounded items require knowledge rarely represented in English-centric datasets.

## 2.4 EVALUATION FRAMEWORK

**Checklist-based Assessment.** Our evaluation methodology centers on detailed checklists that decompose complex answers into specific, measurable criteria. Each problem includes 1–5 evaluation points that assess different aspects of model understanding and reasoning capability. This checklist

Table 1: Performance by category groups. For model families with multiple sizes, only the largest variant is shown. Full results across all model sizes and detailed 13-category breakdowns are in Appendix E. All scores are reported as mean$_{SE}$, where SE is the standard error over 3 independent runs (n=3).

| Model | Entertainment | Scientific | Technical | Daily Life | Overall |
|---|---|---|---|---|---|
| *Proprietary Models* | | | | | |
| Gemini 2.5 Pro | $40.52_{0.61}$ | $51.44_{0.40}$ | $53.89_{0.79}$ | $52.64_{0.93}$ | $48.54_{0.11}$ |
| GPT-5 | $33.07_{0.87}$ | $48.14_{0.96}$ | $55.71_{0.84}$ | $55.98_{0.75}$ | $48.01_{0.19}$ |
| GPT-5 Mini | $27.38_{0.81}$ | $50.62_{0.93}$ | $51.88_{0.74}$ | $51.31_{1.32}$ | $45.21_{0.70}$ |
| Perplexity Sonar-Pro | $32.84_{0.76}$ | $47.98_{0.59}$ | $47.17_{1.23}$ | $49.64_{0.64}$ | $44.28_{0.48}$ |
| Gemini 2.5 Flash | $29.31_{1.09}$ | $45.04_{0.98}$ | $44.05_{0.53}$ | $48.72_{1.38}$ | $41.05_{0.79}$ |
| Grok-4 | $26.88_{0.67}$ | $31.03_{0.64}$ | $44.18_{0.80}$ | $39.67_{0.55}$ | $36.08_{0.30}$ |
| Gemini 2.5 Flash-Lite | $18.39_{0.59}$ | $38.17_{1.47}$ | $32.74_{0.84}$ | $35.47_{0.92}$ | $30.29_{0.24}$ |
| GPT-5 Nano | $11.64_{0.53}$ | $20.10_{1.24}$ | $27.15_{1.36}$ | $29.68_{0.54}$ | $21.22_{0.26}$ |
| *Open-source Models* | | | | | |
| Skywork-R1V3-38B | $15.03_{0.73}$ | $35.31_{0.88}$ | $30.22_{0.49}$ | $33.75_{0.72}$ | $27.76_{0.34}$ |
| Mistral Medium 3.1 | $13.74_{0.80}$ | $30.77_{0.86}$ | $28.87_{0.67}$ | $28.78_{1.01}$ | $24.86_{0.56}$ |
| Gemma-3 27B | $11.59_{0.58}$ | $25.80_{0.61}$ | $22.28_{1.04}$ | $30.85_{0.61}$ | $22.53_{0.16}$ |
| Qwen2.5-VL-72B | $10.89_{0.66}$ | $26.71_{1.49}$ | $21.60_{0.53}$ | $25.61_{0.52}$ | $20.58_{0.46}$ |
| Pixtral Large | $11.43_{0.82}$ | $21.79_{0.50}$ | $21.77_{0.38}$ | $25.65_{0.91}$ | $20.10_{0.24}$ |
| InternVL3.5-38B | $8.81_{0.46}$ | $23.25_{0.61}$ | $17.92_{0.73}$ | $23.36_{0.78}$ | $18.01_{0.22}$ |
| Ovis2-34B | $9.52_{0.47}$ | $21.88_{0.55}$ | $21.00_{0.51}$ | $24.82_{0.58}$ | $18.50_{0.02}$ |
| Mistral Small 24B | $6.46_{0.29}$ | $10.18_{0.45}$ | $13.30_{0.66}$ | $16.20_{0.66}$ | $11.20_{0.01}$ |
| *Korean-specialized Models* | | | | | |
| VARCO-VISION 2.0 (14B) | $7.87_{0.80}$ | $16.56_{0.65}$ | $16.88_{0.57}$ | $22.13_{0.88}$ | $15.55_{0.29}$ |
| HyperCLOVA X-SEED-3B | $6.25_{0.25}$ | $14.87_{0.51}$ | $11.99_{0.50}$ | $17.93_{0.73}$ | $12.66_{0.10}$ |

approach provides several advantages over traditional scoring methods: (1) Fine-grained assessment of partial understanding, (2) Reduced subjectivity through explicit criteria, (3) Diagnostic capability for pinpointing model weaknesses, (4) Scalability for automated evaluation, and (5) Reproducibility and transparency through structured outputs that allow independent re-scoring or cross-judge validation.

**LLM Judge Protocol.** We employ GPT-5-mini as our primary judge, using a structured prompt that enforces consistent scoring across all problems. Each checklist item is scored on a three-level scale: *met* (1.0), *partially met* (0.5), or *not met* (0.0), based on explicit evidence in the model's response. The judge follows strict criteria: completeness statements ("all", "every") require explicit mentions for full credit, method explanations must include concrete steps, and multi-part requirements need at least two distinct examples. Every score is accompanied by supporting evidence, including direct quotes for positive scores or brief explanations for failures, to ensure auditability. The output is returned in a structured format with evidence blocks and fractional scores (e.g., "3.5/5" when two items are partially satisfied and three are fully met). We then parse these outputs programmatically to ensure consistent, reproducible evaluation across all 653 problems and 39 models.

## 3 EXPERIMENTAL SETUP

### 3.1 MODEL SELECTION

We evaluated 39 vision-language models (VLMs) spanning diverse families and scales. This included OpenAI's GPT-5 series (GPT-5, GPT-5-Mini, GPT-5-Nano (OpenAI, 2025)), Google's Gemini (2.5-Pro/Flash/Flash-Lite) and Gemma-3 (27B/12B/4B) (Google DeepMind, 2025; Gemma Team, Google DeepMind, 2025), and proprietary systems on OpenRouter such as Perplexity-Sonar-Pro (Perplexity AI, 2025), xAI-Grok-4 (xAI, 2025), and several Mistral (Medium-3.1, Small-24B) and Pixtral (Large, 12B) models (Mistral AI, 2024; Agrawal et al., 2024). We further incorporated Skywork-R1V3-38B (Shen et al., 2025), AIDC-AI-Ovis2 (34B–1B) (Lu et al., 2025), and InternVL3.5 (38B–1B) (Wang et al., 2025). Finally, we tested Qwen2.5-VL (72B/7B/3B) (Bai et al., 2025), HyperCLOVA-3B (Yoo et al., 2024), and VARCO-VISION-2.0 (14B/1.7B) (NCSOFT AI Center, 2025).

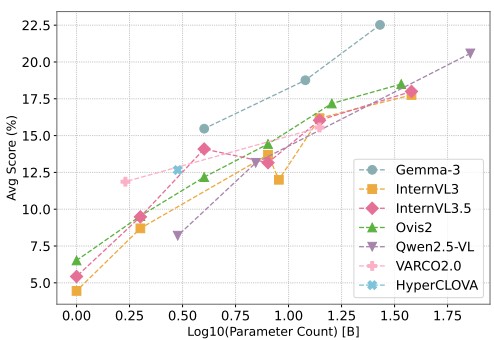 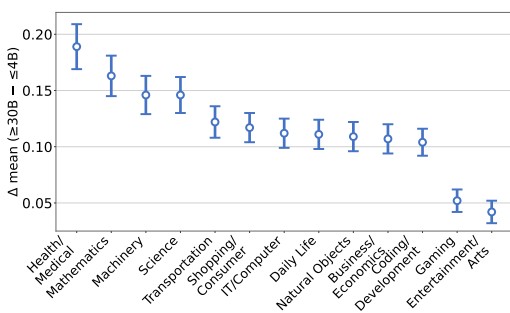

Figure 5: Performance scaling with model parameter count. Accuracy improves up to ~10B parameters but shows diminishing returns thereafter, indicating that benchmark difficulty is not solved by naive scaling.

Figure 6: Domain-level results. Health/Medical shows the highest scores while Entertainment/Gaming remains hardest, confirming that culturally grounded tasks are especially challenging even for large models.

## 3.2 Implementation Details

We standardized decoding parameters across all models using `temperature=0.6`, `top_p=0.95`, and `max_tokens=4096`, and evaluated each question–image pair three times, averaging the scores to reduce variance. Open-weight models with ≤35B parameters were run locally on two NVIDIA H100 GPUs, while larger models (>35B) and proprietary models were accessed via the OpenRouter API with identical decoding parameters to ensure consistency. All model outputs were scored using GPT-5-mini as the evaluation judge with `temperature=1.0`.

## 4 Results and Analysis

### 4.1 Overall Performance

Table 1 summarizes the performance of 39 evaluated VLMs across four major category groups. Even the best-performing models—Gemini 2.5 Pro (48.5%) and GPT-5 (48.0%)—fall short of 50% accuracy, highlighting that authentic, culturally grounded multimodal queries remain far from solved. Proprietary systems consistently outperform open-weight counterparts, with the strongest open-weight models (Skywork-R1V3-38B: 27.8%, Qwen2.5-VL-72B: 25.3%) reaching roughly half the accuracy of top proprietary models. Neither search-augmented models (Perplexity Sonar-Pro: 44.3%) nor reasoning-specialized models (Skywork-R1V3) achieve notable gains, suggesting that solving this benchmark requires capabilities beyond current retrieval-augmented or reasoning-optimized architectures.

Korean-specialized models also struggled to achieve competitive results (VARCO-VISION 2.0 14B: 15.6%, HyperCLOVA X-SEED-3B: 12.7%), indicating that dedicated local models have yet to demonstrate clear advantages on this benchmark. Full per-model and per-domain results are provided in Appendix E.

### 4.2 Performance by Model Scale

Grouping models by size tiers (Small ≤4B, Medium 8–14B, Large ≥30B) reveals a clear monotonic scaling effect. Large models achieve a mean score of 0.3009 (95% CI [0.2974, 0.3046], $n = 31{,}338$), more than double Medium (0.1460) and over triple Small (0.0854). All pairwise differences are statistically significant (permutation $p \approx 0.001$) with substantial effect sizes (Large–Small $\Delta = +0.2155$, Cohen's $d \approx 0.78$). This pattern remains robust even when aggregating across runs (Large 0.2030 vs Small 0.0854), confirming that the scale effect is not an artifact of evaluation variance.

At the family level, commercial systems (Gemini, GPT, Sonar) consistently outperform open-weight models (e.g., InternVL3), with effect sizes in the $d = 0.7$–$1.2$ range (e.g., Gemini-2.5-Pro vs In-

Table 2: Checklist rule statistics showing binary score distribution. The scores are averaged across models and domains.

| Rule | p(0.0) | p(1.0) | avg_met | 95% CI |
|---|---|---|---|---|
| Explicitness | 92.1% | 7.9% | 0.079 | 0.075–0.084 |
| Variety | 90.1% | 9.9% | 0.099 | 0.089–0.109 |
| Unknown | 89.1% | 10.9% | 0.109 | 0.104–0.115 |
| Synonym | 88.0% | 12.0% | 0.120 | 0.097–0.143 |
| Method | 87.4% | 12.6% | 0.126 | 0.122–0.130 |
| Completeness | 87.4% | 12.6% | 0.126 | 0.116–0.136 |

ternVL3 $\Delta \approx 0.49$, $d \approx 1.21$). The plateau beyond $\sim 10B$ parameters (Figure 5) highlights that model size alone is insufficient to close the performance gap, suggesting that improvements in reasoning and cultural grounding are needed rather than just scaling.

### 4.3 PERFORMANCE BY DOMAIN

Performance varies widely across the 13 domains (global mean = 0.1987, range 0.1179–0.332). Health/Medical achieves the highest checklist satisfaction (0.332), followed by Science (0.250), while Entertainment/Arts (0.118) and Gaming (0.119) remain the most challenging. Within all domains, large models ($\geq 30B$) consistently outperform small models ($\leq 4B$) (permutation $p < 0.05$), with the largest gains in Health/Medical ($\Delta = +0.189$) and Mathematics ($\Delta = +0.163$). Even in Gaming and Entertainment, scale effects remain positive though absolute performance stays low (Figure 6).

### 4.4 ERROR ANALYSIS

We analyzed 59k checklist items across six rule types: EXPLICITNESS, VARIETY, METHOD, COMPLETENESS, SYNONYM, and UNKNOWN. Table 2 shows that failures concentrate in EXPLICITNESS (92.1% unmet) and VARIETY (90.1%), indicating that models often omit key terms and fail to enumerate required items. METHOD and COMPLETENESS fare slightly better (12.6% success each) but still expose major gaps in procedural reasoning. Partial credit (0.5) was assigned in under 1% of cases, so scoring was effectively binary.

Illustrative failure cases include:

- EXPLICITNESS: In medical queries requiring "side effects," models described symptoms but omitted the term.
- VARIETY: In economics tasks asking for multiple risks, models listed only one.
- METHOD: In scientific problems, models gave only the final answer without steps.
- COMPLETENESS: In transit comparisons, answers covered system A but omitted system B.
- SYNONYM: In consumer queries, models failed to equate "sale" with "discount."

We also observe a strong correlation between EXPLICITNESS and METHOD failures ($r = 0.73$), suggesting that vague responses systematically lack procedural explanations. These patterns are especially common in entertainment and gaming, where contextual reasoning is critical. Overall, error analysis highlights that current VLMs struggle most with explicitness and multi-step coverage. While larger models mitigate these issues somewhat, high failure rates persist, showing that cultural and procedural reasoning remain unresolved beyond parameter scaling.

### 4.5 EFFECT OF SEARCH-AUGMENTED INFERENCE

We evaluated whether enabling online access improves performance by comparing three models with and without web search capabilities. GPT-5 used its native browsing tool, while Mistral-Medium-3.1 and Qwen2.5-VL-72B-Instruct accessed the Exa API.

Results (Table 3) show no consistent benefit: Mistral sees moderate gains, but GPT-5 and Qwen2.5-VL actually perform worse. This can be explained by two limiting factors: language bias, since

Table 3: Performance with and without online access. Gains are inconsistent, highlighting limitations of current web search for Korean multimodal queries.

| Model | Offline | Online | $\Delta$ |
|---|---|---|---|
| GPT-5 | 48.01 | 46.25 | $-1.76$ |
| Mistral-Medium-3.1 | 24.86 | 31.47 | $+6.61$ |
| Qwen2.5-VL-72B-Instruct | 25.31 | 17.24 | $-8.07$ |

Table 4: Inter-judge agreement across four LLM judges. Values show pairwise Pearson correlations (all $> 0.86$); Spearman correlations range from $0.866$ to $0.901$. Overall agreement across all judges yields Krippendorff's $\alpha = 0.867$.

| | GPT-5-mini | GPT-5 | Gemini-2.5-Pro | Gemini-2.5-Flash |
|---|---|---|---|---|
| GPT-5-mini | 1.000 | 0.868 | 0.900 | 0.903 |
| GPT-5 | 0.868 | 1.000 | 0.897 | 0.863 |
| Gemini-2.5-Pro | 0.900 | 0.897 | 1.000 | 0.887 |
| Gemini-2.5-Flash | 0.903 | 0.863 | 0.887 | 1.000 |
| **Krippendorff's** $\alpha = 0.867$ | | | | |

Korean-relevant webpages are rarely surfaced by current search engines, and the lack of image-aware retrieval, as search tools cannot incorporate visual context and leave many queries under-specified. These findings suggest that **HAERAE-Vision** is robust to contamination from search-augmented models, and as web search improves, it can serve as a framework for tracking progress in both VLM capabilities and multimodal retrieval systems.

### 4.6 VALIDATION AND RELIABILITY

We assessed evaluation reliability by measuring inter-judge agreement among four LLM judges (GPT-5, GPT-5-mini, Gemini-2.5-Pro, Gemini-2.5-Flash). A stratified random sample of 250 model responses (50 per 0.2-score interval) was re-evaluated under identical protocols. Table 4 shows consistently high correlations, with Pearson ranging from 0.863 to 0.903 and Spearman from 0.866 to 0.901. Krippendorff's $\alpha = 0.867$ exceeds the conventional 0.80 threshold, indicating substantial agreement across models with different architectures.

We also measured alignment with human judgments using the same 250-sample dataset. Four independent annotators (non-authors) rated GPT-5-mini's scores on a 5-point appropriateness scale (5 = very appropriate, 1 = very inappropriate), with each response reviewed by two annotators. The mean appropriateness score was $4.13$ (SD = 1.23), with 73.2% of ratings in the 4–5 range. Inter-annotator agreement was substantial (Cohen's $\kappa = 0.493$, Pearson $r = 0.673$, Spearman $\rho = 0.713$), and $\pm 1$-point agreement reached 98.4%. These results demonstrate that our LLM-as-a-judge setup provides a consistent and reproducible evaluation signal, comparable to established benchmarks (Zheng et al., 2023; Liu et al., 2023). Analysis of low-rated cases shows most failures involved superficial keyword matching or excessive leniency (examples in G).

## 5 DISCUSSION

### 5.1 CHECKLIST: BEYOND SIMPLE CORRECTNESS

Our checklist-based evaluation enforces strict standards that require comprehensive, context-aware explanations rather than simple answer matching, reflecting authentic user expectations in Korean online communities where brief correctness alone rarely satisfies information needs. The framework penalizes superficial responses—for example, simply stating that an insect is a "silverfish" earns no or partial credit, whereas full credit requires describing its appearance, explaining its habitat and behavior, and addressing safety implications. This design mirrors the detailed, context-rich answers valued by real users. Our error analysis supports this approach: models frequently produce correct

but incomplete answers, with 92.1% failing explicitness requirements, highlighting a persistent gap between current VLM outputs and real-world expectations for depth and clarity.

## 5.2 LIMITATIONS

Our work has several limitations. First, the benchmark is limited to Korean language and cultural context, though the methodology is designed to generalize to other populations. Second, while the six-stage pipeline ensures high data quality, the stringent 0.76% survival rate may risk excluding rare but informative edge cases. Third, checklist-based evaluation relies on predefined criteria that, while comprehensive, may not capture every nuance of response quality.

## 5.3 ETHICS AND DATA GOVERNANCE

All data are sourced from public Korean community posts under each site's terms of use. Sensitive content is filtered, and personally identifiable information is removed or blurred. We release a balanced 20% development subset covering 12 categories, while the Health/Medical category is withheld due to privacy constraints. The full 13-category test set is hosted on a rate-limited, anonymous evaluation server to prevent overfitting and support fair model comparison.

## 6 RELATED WORK

Recent work has highlighted the importance of cultural context in VLM evaluation. CulturalVQA (Nayak et al., 2024) revealed that current models are strongly biased toward high-resource cultural settings, underscoring the need for culturally grounded benchmarks.

In the Korean context, several efforts have advanced cultural integration. K-Viscuit (Park et al., 2024) emphasizes human-validated, culture-centric multiple-choice VQA, KOFFVQA (Kim & Jung, 2025) introduces free-form Korean VQA, and Ko-PIQA (Choi et al., 2025) targets Korean commonsense and culturally grounded plausibility judgments beyond surface cues. Community-authentic datasets such as VQAonline (Chen et al., 2024) further demonstrate the value of sourcing real user queries for multimodal evaluation. Nevertheless, these lines of work still leave gaps in capturing the messy, under-specified multimodal question patterns common in user communities.

Concurrently, benchmark research has shifted toward quality-focused evaluation. VLRMBench emphasizes systematic error analysis and fine-grained, step-level signals for diagnosing model failures (Ruan et al., 2025), while LIMO shows that carefully curated, high-fidelity examples can rival or outperform much larger datasets (Ye et al., 2025). These trends motivate our checklist-based scoring with LLM judges and our rigorous six-stage curation pipeline.

Our work integrates these directions by combining cultural grounding, Korean specialization, and rigorous data curation. We collect authentic multimodal queries from diverse Korean online communities and filter them through a six-stage pipeline with consensus-based difficulty filtering and multiphase validation. This yields a challenging, high-fidelity benchmark with only 0.76% of initial data surviving, establishing what we believe to be a principled, replicable framework for cross-cultural multimodal evaluation that can be extended to other languages and populations.

## 7 CONCLUSION

We introduce **HAERAE-Vision**, a Korean vision–language benchmark distilled from 86,052 community questions into 653 rigorously validated items via a six-stage pipeline. The benchmark stress-tests current systems, with top performers below 50% accuracy and recurring failures in explicitness and procedural reasoning. Our contributions are a reproducible construction recipe for authentic multimodal items, culturally grounded tasks, and a checklist-based evaluation with reliable LLM judging. We release a balanced development subset and host the full test set on a rate-limited server to enable fair comparison across models.

ETHICS STATEMENT

We used ChatGPT and Claude to refine writing and assist with code scaffolding. All experiments and analyses were independently designed and verified by the authors.

REPRODUCIBILITY STATEMENT

We release evaluation prompts, model outputs, and analysis code at `https://anonymous.4open.science/r/haerae-bench-vision-AA34` to ensure full reproducibility of our results.

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
