## A  DETAILED PLATFORM DESCRIPTIONS

We collected data from nine Korean online platforms representing diverse user communities and domain expertise. Table 1 provides detailed information about each platform. These platforms were selected to ensure comprehensive coverage of different user demographics, expertise levels, and domain-specific knowledge, reflecting the diversity of real-world multimodal questions Korean users encounter online.

Table 1: Korean online platforms used for data collection

| Platform | Category | Description |
|---|---|---|
| Naver KnowledgeIn | General Q&A | Korea's largest general Q&A platform covering everyday queries, academic subjects, and technical issues |
| BRIC | Science Community | Specialized community for biological research and biotechnology with scientific discussions and professional knowledge sharing |
| Ruliweb | Gaming Community | Major gaming community covering video games, hardware reviews, game mechanics, and technical gaming issues |
| MonsterZym | Fitness Community | Fitness and bodybuilding community discussing workout routines, nutrition, supplements, and exercise techniques |
| Quasarzone | Hardware Community | Hardware enthusiast community focused on computer components, electronics, PC building, and technology reviews |
| i-Boss | Business Platform | Business and entrepreneurship platform for startup strategies, operations, marketing, and professional development |
| Inflearn | Coding Education | Online learning platform with community features for programming questions and coding experiences |
| Codeit | Coding Education | Coding education platform with forums for programming discussions and technical support |
| Okky | Developer Community | Developer community platform for programming discussions, career advice, and technical problem-solving |

## B  STAGE 2 PROMPT EXCERPTS

We used three LLM-based filters in Stage 2: content safety, objectivity, and temporal dependency. Below we excerpt only the core exclusion criteria from the prompts (full wording omitted).

### B.1  CONTENT SAFETY

Mark as inappropriate if the question–image pair includes:
- Political content (politicians, parties, elections, political opinions)
- Religious advocacy/criticism or conflicts
- Hate/discrimination
- Suicide or self-harm; sensitive mental-health topics
- Sexual/adult content, nudity, explicit innuendo

### B.2  OBJECTIVITY

Mark as inappropriate if the pair is subjective or ambiguous, e.g.:
- Preference/aesthetic judgments ("pretty/ugly", "which outfit is nicer?")
- Suitability/personal advice without criteria
- Moral/intentionality speculation ("who is wrong?", "good person?")
- Multiple valid interpretations or unverifiable answers

| Game (Stardew Valley) | Economics/Management |
|---|---|
| *"What is the circled item in the screenshot?"* | *"Cost allocation: is S2 missing 100,000?"* |
| • Identify circled item as a sap tap (수액채취기) | • Provide correct S1/S2 values |
| • Mention install only on fully grown trees | • Reset self-allocation entries to zero |
| • Explain how to obtain/craft it | • Derive allocation ratios (0.5F, 0.4M) |
| • Note sap can be collected after time | |

| Daily Life | Science |
|---|---|
| *"Is this ceiling tile asbestos?"* | *"Why does neutron mass ratio decrease?"* |
| • Identify material as gypsum, not asbestos | • Explain neutron beta decay |
| • Explain gypsum board contains no asbestos | • Clarify neutrons inside He nucleus |
| • Explicitly name "석고텍스" | • Relate $x$-axis to cosmic cooling |
| • Assure user it is safe | • Interpret H:He ratio $\approx 3:1$ |

Figure 1: Examples of checklist decomposition across domains, generated in Stage 5. For brevity, the checklists shown here are abbreviated; full checklists typically contain 1–5 criteria per item.

## B.3 TEMPORAL DEPENDENCY

Mark as inappropriate if the pair requires time-specific information, e.g.:
- "today/now" weather, traffic, store hours, last train
- Current events or status queries ("is it open now?", "stock price today?")
- Questions that become invalid/meaningless as time passes

## STAGE 4 PROMPT EXCERPT (IMAGE DEPENDENCY RUBRIC)

**Input:** (Q), model answer with image, model answer without image, optional gold answer snippet.
**Task:** Compare the two answers and decide image dependency.
**Decision labels**
- IMAGE_REQUIRED: with-image answer is substantially more accurate/specific; text-only answer is vague, incorrect, or explicitly requests the image.
- NO_IMAGE_NEEDED: both answers are comparable in correctness and specificity without relying on visual cues.
- UNCERTAIN: evidence is inconclusive (e.g., partial improvements or conflicting signals).

**Scoring (1–5 quality gap)**
- 1: negligible difference; 3: clear but moderate gain; 5: decisive gain (critical visual details).

**Output (natural language)**
- `Judgment`: IMAGE_REQUIRED / NO_IMAGE_NEEDED / UNCERTAIN
- `Reason`: brief comparison citing concrete differences
- `QualityGap`: integer in {1,2,3,4,5}

## C STAGE 5 PROMPTS AND CHECKLIST EXAMPLES

This appendix provides the instruction prompt used for checklist generation along with illustrative examples of the resulting decompositions. We used GPT-4-mini to derive structured criteria directly from reference answers that users found satisfactory. These checklists therefore represent strict, human-aligned evaluation standards: a model must satisfy all listed criteria to be considered correct.

# D  PLATFORM-WISE FILTERING STATISTICS

Table 2 provides a detailed breakdown of data collection and filtering across all platforms.

Table 2: Detailed data collection and filtering statistics by platform (Stages 1–6). Coding platforms include Inflearn, Codeit, and Okky combined.

| Platform | Raw Data | Appropri. | Difficulty | Image Dep. | Human Val. | Final | Survival |
|---|---|---|---|---|---|---|---|
| KnowledgeIn | 31,484 | 10,495 | 1,404 | 648 | 441 | 441 | 1.4% |
| BRIC | 291 | 291 | 163 | 60 | 42 | 42 | 14.4% |
| Ruliweb | 305 | 240 | 54 | 42 | 32 | 32 | 10.5% |
| Coding | 27,896 | 8,369 | 837 | 198 | 135 | 135 | 0.5% |
| MonsterZym | 3,090 | 3,090 | 2,234 | 8 | 6 | 6 | 0.2% |
| Quasarzone | 2,986 | 896 | 90 | 22 | 15 | 15 | 0.5% |
| i-Boss | 20,000 | 20,000 | 578 | 62 | 42 | 42 | 0.2% |
| Total | 86,052 | 43,381 | 5,360 | 1,040 | 713 | 653 | 0.76% |

# E  INVESTIGATING FAILURE MODES

In Table 3, we observe that VARCO-VISION and HyperCLOVA X—two Korean-focused VLMs—underperform multilingual counterparts of similar scale. While the precise reasons remain unclear due to the closed nature of these models and limited information about their training, we propose two possible explanations:

(A) **Training Data Coverage.**  Current benchmarks that capture progress on culturally grounded, information-deficient queries are scarce.  Model developers may not have explicitly emphasized such aspects in their training data, leading to weaker performance on this type of evaluation.

(B) **Pretraining Scale and Robustness.** Robustness to imperfect or fragmented user queries may emerge from exposure to large-scale, diverse pretraining corpora. Larger multilingual models are more likely to encounter noisy, colloquial, or partially specified inputs, thereby preparing them better for benchmarks of this kind.

Table 3: Complete performance across all 13 categories for all evaluated models (scores in %). All scores are reported as mean$_{SE}$, where SE is the standard error over 3 independent runs (n=3).

| Model | IT | Health | Game | Econ | Sci | Mach | Daily | Shop | Math | Ent | Trans | Nature | Code | Avg |
|---|---|---|---|---|---|---|---|---|---|---|---|---|---|---|
| *Proprietary Models* | | | | | | | | | | | | | | |
| Gemini 2.5 Pro | $50.73_{1.63}$ | $62.17_{2.33}$ | $36.67_{1.75}$ | $51.09_{1.72}$ | $39.93_{1.43}$ | $56.22_{4.32}$ | $51.32_{2.57}$ | $46.00_{5.15}$ | $60.94_{1.36}$ | $44.37_{1.18}$ | $57.69_{2.81}$ | $53.45_{0.57}$ | $50.91_{0.92}$ | $48.54_{0.18}$ |
| Gemini 2.5 Flash | $42.98_{0.34}$ | $56.10_{3.95}$ | $26.70_{1.04}$ | $48.05_{6.64}$ | $39.62_{3.14}$ | $45.86_{1.70}$ | $44.59_{2.10}$ | $46.13_{5.19}$ | $51.14_{3.87}$ | $31.92_{3.63}$ | $48.12_{2.37}$ | $44.37_{0.99}$ | $39.26_{2.25}$ | $41.05_{1.38}$ |
| Gemini 2.5 Flash Lite | $25.92_{1.82}$ | $43.54_{4.48}$ | $17.97_{1.92}$ | $38.84_{3.82}$ | $41.73_{1.47}$ | $38.30_{3.10}$ | $30.67_{0.88}$ | $28.82_{2.38}$ | $45.62_{7.49}$ | $18.82_{0.68}$ | $34.10_{3.78}$ | $27.16_{0.32}$ | $32.63_{2.66}$ | $30.29_{0.42}$ |
| GPT 5 | $59.95_{2.01}$ | $62.61_{2.59}$ | $32.34_{2.08}$ | $58.41_{1.63}$ | $36.31_{1.60}$ | $52.85_{4.72}$ | $46.93_{2.83}$ | $55.96_{3.14}$ | $54.70_{4.54}$ | $33.80_{2.20}$ | $54.97_{2.43}$ | $53.42_{1.23}$ | $55.07_{1.24}$ | $48.01_{0.32}$ |
| GPT 5 Mini | $49.59_{3.74}$ | $60.45_{2.40}$ | $29.22_{1.71}$ | $50.19_{5.53}$ | $52.49_{0.44}$ | $51.68_{1.47}$ | $50.28_{4.75}$ | $44.33_{4.96}$ | $58.19_{3.94}$ | $25.54_{2.20}$ | $49.23_{3.11}$ | $41.17_{2.73}$ | $57.02_{0.53}$ | $45.21_{1.21}$ |
| GPT 5 Nano | $22.99_{2.64}$ | $45.98_{1.02}$ | $10.46_{1.71}$ | $24.81_{0.65}$ | $11.47_{1.21}$ | $26.59_{7.45}$ | $21.49_{1.41}$ | $26.42_{3.26}$ | $23.56_{2.12}$ | $12.81_{0.67}$ | $26.17_{1.63}$ | $25.27_{1.60}$ | $32.84_{4.80}$ | $21.22_{0.46}$ |
| Grok 4 | $39.64_{1.89}$ | $36.96_{1.16}$ | $29.00_{1.49}$ | $44.44_{2.79}$ | $40.70_{1.13}$ | $47.63_{1.86}$ | $40.57_{1.60}$ | $36.73_{1.65}$ | $22.09_{2.86}$ | $24.77_{1.78}$ | $50.43_{4.90}$ | $30.29_{1.28}$ | $39.02_{0.20}$ | $36.08_{0.53}$ |
| *Open-source Models* | | | | | | | | | | | | | | |
| *Mistral/Pixtral Family* | | | | | | | | | | | | | | |
| Mistral Medium 3.1 | $24.77_{2.76}$ | $37.01_{5.96}$ | $16.01_{1.49}$ | $28.48_{2.48}$ | $33.70_{1.23}$ | $34.14_{1.20}$ | $24.41_{1.06}$ | $25.22_{2.51}$ | $38.99_{3.41}$ | $11.46_{2.32}$ | $25.46_{2.65}$ | $19.62_{2.62}$ | $31.09_{2.35}$ | $24.86_{0.98}$ |
| Pixtral Large | $19.09_{1.82}$ | $35.09_{2.33}$ | $11.33_{1.33}$ | $24.32_{3.36}$ | $27.40_{2.29}$ | $23.41_{1.21}$ | $24.16_{1.09}$ | $19.01_{4.66}$ | $19.89_{1.10}$ | $11.54_{2.50}$ | $21.93_{1.34}$ | $18.08_{0.62}$ | $22.64_{0.67}$ | $20.10_{0.41}$ |
| Mistral Small 24B | $15.38_{1.93}$ | $25.07_{4.25}$ | $7.00_{1.35}$ | $22.29_{1.84}$ | $20.47_{1.46}$ | $21.53_{2.01}$ | $13.07_{2.67}$ | $15.34_{3.68}$ | $18.57_{1.81}$ | $7.76_{1.09}$ | $13.84_{3.49}$ | $10.61_{1.77}$ | $16.36_{2.46}$ | $14.43_{0.41}$ |
| Pixtral 12B | $8.76_{0.77}$ | $24.19_{3.58}$ | $6.74_{0.94}$ | $17.49_{0.53}$ | $14.12_{0.11}$ | $16.46_{2.65}$ | $11.66_{2.40}$ | $11.44_{1.34}$ | $6.83_{2.27}$ | $6.17_{0.35}$ | $15.06_{2.39}$ | $9.60_{0.46}$ | $12.94_{2.80}$ | $11.20_{0.02}$ |
| *Google Gemma Family* | | | | | | | | | | | | | | |
| Gemma 3 27B | $20.31_{1.18}$ | $40.90_{1.49}$ | $13.75_{1.55}$ | $31.71_{3.21}$ | $34.93_{1.52}$ | $27.36_{6.28}$ | $26.72_{1.12}$ | $24.07_{2.01}$ | $23.85_{2.74}$ | $9.43_{1.30}$ | $20.66_{2.62}$ | $18.61_{0.40}$ | $20.81_{2.15}$ | $22.53_{0.28}$ |
| Gemma 3 12B | $15.15_{0.69}$ | $36.60_{1.32}$ | $10.52_{1.44}$ | $27.91_{1.30}$ | $28.79_{1.39}$ | $27.44_{3.60}$ | $19.20_{1.27}$ | $22.40_{1.47}$ | $17.25_{2.89}$ | $7.23_{1.12}$ | $21.01_{1.65}$ | $13.43_{1.47}$ | $23.41_{0.13}$ | $18.76_{0.63}$ |
| Gemma 3 4B | $12.43_{1.63}$ | $34.23_{1.08}$ | $8.91_{0.96}$ | $19.67_{4.37}$ | $22.50_{0.12}$ | $21.25_{1.33}$ | $15.59_{0.87}$ | $18.21_{1.21}$ | $13.54_{2.63}$ | $6.84_{1.10}$ | $19.56_{2.12}$ | $14.68_{1.08}$ | $13.45_{0.88}$ | $15.47_{0.78}$ |
| *AIDC-AI Ovis2 Family* | | | | | | | | | | | | | | |
| Ovis2-34B | $15.90_{1.35}$ | $40.15_{2.16}$ | $9.87_{0.77}$ | $19.44_{0.45}$ | $23.97_{0.56}$ | $29.46_{1.47}$ | $19.43_{0.58}$ | $20.27_{3.31}$ | $22.91_{2.46}$ | $9.18_{1.41}$ | $21.86_{2.89}$ | $18.77_{1.37}$ | $16.78_{0.26}$ | $18.50_{0.03}$ |
| Ovis2-16B | $11.20_{1.67}$ | $38.98_{0.75}$ | $8.08_{0.18}$ | $21.58_{1.27}$ | $24.68_{0.80}$ | $23.94_{3.50}$ | $21.20_{3.52}$ | $14.83_{3.00}$ | $24.32_{1.31}$ | $8.72_{1.57}$ | $20.21_{0.84}$ | $16.47_{0.63}$ | $16.12_{1.92}$ | $17.18_{0.50}$ |
| Ovis2-8B | $9.80_{0.30}$ | $33.62_{1.54}$ | $6.07_{0.30}$ | $19.18_{3.28}$ | $19.45_{1.85}$ | $21.02_{1.98}$ | $18.37_{1.83}$ | $13.51_{1.33}$ | $19.81_{5.29}$ | $8.04_{0.53}$ | $17.42_{3.08}$ | $13.17_{0.35}$ | $14.77_{1.80}$ | $14.46_{0.37}$ |
| Ovis2-4B | $6.76_{1.75}$ | $23.66_{3.93}$ | $6.00_{0.27}$ | $15.89_{2.76}$ | $16.16_{1.17}$ | $17.05_{3.15}$ | $16.43_{1.51}$ | $10.68_{2.89}$ | $13.16_{0.84}$ | $7.28_{0.50}$ | $17.65_{3.01}$ | $14.26_{0.58}$ | $8.31_{1.00}$ | $12.18_{0.11}$ |
| Ovis2-2B | $6.14_{0.22}$ | $16.10_{1.01}$ | $5.30_{0.83}$ | $13.74_{2.34}$ | $12.24_{1.70}$ | $13.64_{4.43}$ | $11.99_{1.14}$ | $11.27_{2.01}$ | $6.57_{1.32}$ | $7.28_{0.64}$ | $11.33_{2.19}$ | $9.73_{0.56}$ | $8.98_{3.88}$ | $9.54_{0.22}$ |
| Ovis2-1B | $4.83_{0.91}$ | $12.62_{2.58}$ | $4.74_{0.31}$ | $8.07_{1.07}$ | $7.52_{0.71}$ | $5.95_{1.12}$ | $8.03_{0.98}$ | $8.11_{1.97}$ | $6.57_{2.40}$ | $5.05_{0.98}$ | $8.10_{2.55}$ | $6.80_{1.38}$ | $4.43_{1.13}$ | $6.52_{0.25}$ |
| *OpenGVLab InternVL3.5 Family* | | | | | | | | | | | | | | |
| InternVL3.5 38B | $14.94_{0.63}$ | $30.95_{4.82}$ | $9.09_{1.57}$ | $24.85_{1.52}$ | $28.79_{0.27}$ | $20.90_{4.44}$ | $19.25_{1.90}$ | $18.40_{0.19}$ | $24.54_{2.47}$ | $8.53_{0.17}$ | $21.10_{0.84}$ | $16.41_{1.98}$ | $14.76_{2.12}$ | $18.01_{0.39}$ |
| InternVL3.5 14B | $15.50_{2.05}$ | $26.81_{4.46}$ | $8.26_{1.12}$ | $20.72_{0.96}$ | $24.64_{1.18}$ | $17.41_{3.95}$ | $14.67_{1.98}$ | $17.70_{3.09}$ | $26.45_{1.63}$ | $7.74_{0.53}$ | $15.76_{1.58}$ | $12.05_{1.07}$ | $19.72_{3.13}$ | $16.04_{0.37}$ |
| InternVL3.5 8B | $10.22_{1.08}$ | $23.11_{3.12}$ | $7.14_{0.57}$ | $20.44_{1.87}$ | $20.14_{1.89}$ | $16.16_{3.35}$ | $11.27_{1.56}$ | $11.99_{2.92}$ | $22.96_{2.08}$ | $5.29_{0.79}$ | $12.68_{1.73}$ | $12.57_{0.25}$ | $13.01_{1.22}$ | $13.16_{0.82}$ |
| InternVL3.5 4B | $7.70_{1.15}$ | $23.33_{0.30}$ | $7.72_{0.52}$ | $19.71_{1.60}$ | $23.20_{1.24}$ | $18.84_{0.42}$ | $15.11_{1.52}$ | $14.98_{2.03}$ | $25.72_{2.64}$ | $6.48_{0.96}$ | $13.83_{1.36}$ | $11.78_{1.37}$ | $14.90_{2.25}$ | $14.09_{0.28}$ |
| InternVL3.5 2B | $5.32_{0.25}$ | $20.86_{4.13}$ | $5.24_{0.34}$ | $15.50_{1.86}$ | $16.05_{0.87}$ | $12.69_{2.87}$ | $8.94_{1.54}$ | $7.69_{1.60}$ | $14.18_{1.71}$ | $5.63_{1.26}$ | $10.14_{3.14}$ | $7.03_{0.51}$ | $9.07_{2.28}$ | $9.48_{0.49}$ |
| InternVL3.5 1B | $3.21_{0.43}$ | $7.94_{2.99}$ | $3.39_{0.09}$ | $10.32_{0.07}$ | $9.12_{0.32}$ | $5.74_{0.58}$ | $3.29_{1.02}$ | $7.79_{1.30}$ | $10.22_{1.53}$ | $3.24_{0.57}$ | $7.31_{1.05}$ | $2.93_{0.74}$ | $5.64_{0.43}$ | $5.43_{0.13}$ |
| *Qwen Family* | | | | | | | | | | | | | | |
| Qwen2.5 VL 72B | $16.53_{1.36}$ | $31.30_{1.38}$ | $11.80_{2.24}$ | $25.55_{1.06}$ | $28.46_{2.62}$ | $23.55_{1.42}$ | $19.72_{0.38}$ | $25.86_{3.14}$ | $32.32_{7.22}$ | $9.97_{0.45}$ | $21.02_{2.62}$ | $19.36_{0.79}$ | $25.31_{1.59}$ | $20.58_{0.80}$ |
| Qwen2.5 VL 7B | $10.33_{0.70}$ | $21.04_{4.51}$ | $5.95_{1.26}$ | $18.96_{1.05}$ | $20.49_{3.89}$ | $18.50_{3.79}$ | $13.70_{0.92}$ | $17.00_{4.00}$ | $13.26_{4.07}$ | $6.71_{0.28}$ | $14.06_{1.66}$ | $12.35_{0.74}$ | $13.28_{2.86}$ | $13.15_{0.86}$ |
| Qwen2.5 VL 3B | $6.08_{2.15}$ | $18.49_{3.90}$ | $2.82_{0.44}$ | $12.76_{1.17}$ | $11.54_{1.70}$ | $13.76_{2.51}$ | $9.22_{0.16}$ | $6.89_{1.47}$ | $10.14_{0.98}$ | $4.88_{0.18}$ | $10.31_{3.46}$ | $7.85_{0.38}$ | $6.54_{0.84}$ | $8.20_{0.36}$ |
| *Other Open-source* | | | | | | | | | | | | | | |
| Skywork-R1V3-38B | $27.12_{0.74}$ | $47.94_{2.92}$ | $15.30_{1.63}$ | $32.37_{2.44}$ | $36.84_{0.69}$ | $37.25_{1.80}$ | $26.43_{2.63}$ | $28.27_{1.95}$ | $41.71_{4.53}$ | $14.76_{1.96}$ | $30.10_{2.73}$ | $27.38_{0.26}$ | $26.42_{0.26}$ | $27.76_{0.58}$ |
| *Korean-specialized Models* | | | | | | | | | | | | | | |
| VARCO-VISION-2.0-14B | $11.90_{0.79}$ | $34.76_{4.78}$ | $7.94_{0.85}$ | $17.83_{2.30}$ | $22.03_{2.71}$ | $23.46_{3.16}$ | $21.89_{1.09}$ | $14.05_{2.80}$ | $12.68_{1.90}$ | $7.80_{2.64}$ | $18.84_{1.75}$ | $14.97_{0.57}$ | $13.31_{1.31}$ | $15.55_{0.50}$ |
| HyperCLOVA-3B | $8.42_{0.98}$ | $29.74_{2.98}$ | $6.33_{0.49}$ | $15.17_{1.40}$ | $18.54_{0.41}$ | $15.80_{2.14}$ | $13.38_{0.67}$ | $13.43_{3.83}$ | $9.86_{2.44}$ | $6.16_{0.70}$ | $14.20_{1.75}$ | $16.21_{0.93}$ | $9.53_{1.86}$ | $12.66_{0.18}$ |
| VARCO-VISION-2.0-1.7B | $8.09_{1.21}$ | $21.34_{1.50}$ | $5.95_{2.38}$ | $16.07_{0.96}$ | $17.79_{0.63}$ | $16.22_{1.16}$ | $12.70_{0.32}$ | $12.88_{1.08}$ | $12.54_{5.35}$ | $8.11_{0.68}$ | $12.81_{1.01}$ | $12.13_{0.72}$ | $10.46_{3.57}$ | $11.87_{0.46}$ |

# F  ANNOTATION GUIDELINES

Seven Korean-speaking annotators conducted human validation using a custom web-based tool (Figure 2). Annotators were provided with comprehensive guidelines covering five evaluation dimensions:

## F.1  CORE EVALUATION CRITERIA

**Image-Question Relevance**: Assess whether images provide essential visual information required to answer the question. Images should contain specific visual elements that cannot be determined from text alone.

**Question-Answer Quality**: Evaluate question clarity, answerability, and reference answer accuracy. Questions should be unambiguous with verifiable answers, while reference answers should be comprehensive and correct.

**Checklist Validation**: Review each checklist item for necessity, clarity, and completeness. Items should capture essential answer components using unambiguous, measurable criteria that together represent the full scope of required understanding.

**Category Appropriateness**: Verify correct classification into one of 13 domain categories based on primary subject matter and required expertise.

**Overall Assessment**: Flag items with fundamental issues such as inappropriate content, cultural insensitivity, or unsolvable questions.

## F.2  QUALITY STANDARDS

Annotators applied conservative filtering criteria, removing any item flagged as problematic to maintain high dataset quality. Specific attention was given to:

- Cultural specificity and Korean context requirements
- Image dependency verification through visual inspection
- Checklist comprehensiveness and granularity
- Answer completeness relative to community expectations

The annotation process employed automatic progress saving and session management to ensure data integrity and allow work resumption across multiple sessions.

# G  HUMAN EVALUATION FEEDBACK ANALYSIS

Table 4 presents representative examples of human annotator feedback for inappropriate judge evaluations, revealing systematic failure patterns.

| Rating | Human Reasoning (translated) |
|---|---|
| Very Inappropriate | "Judge awarded points based on superficial word matching rather than actual checklist compliance" |
| Inappropriate | "Judge gave 1 point despite response not addressing checklist criteria, incorrectly interpreting explicit mention as meeting requirements" |
| Inappropriate | "Checklists 1,2,4 satisfied. Item 3 not clearly inappropriate but ambiguous and open to interpretation" |
| Inappropriate | "Even if intent aligns with checklist, response lacks clarity and remains ambiguous" |
| Inappropriate | "Judge overlooked insufficient explanations that clearly failed checklist requirements" |

Table 4: Representative human feedback explaining inappropriate judge ratings.

Analysis reveals judge failures primarily stem from: (1) superficial keyword matching without semantic understanding, (2) excessive leniency toward incomplete responses, and (3) difficulty distinguishing between implicit intent and explicit satisfaction of requirements.

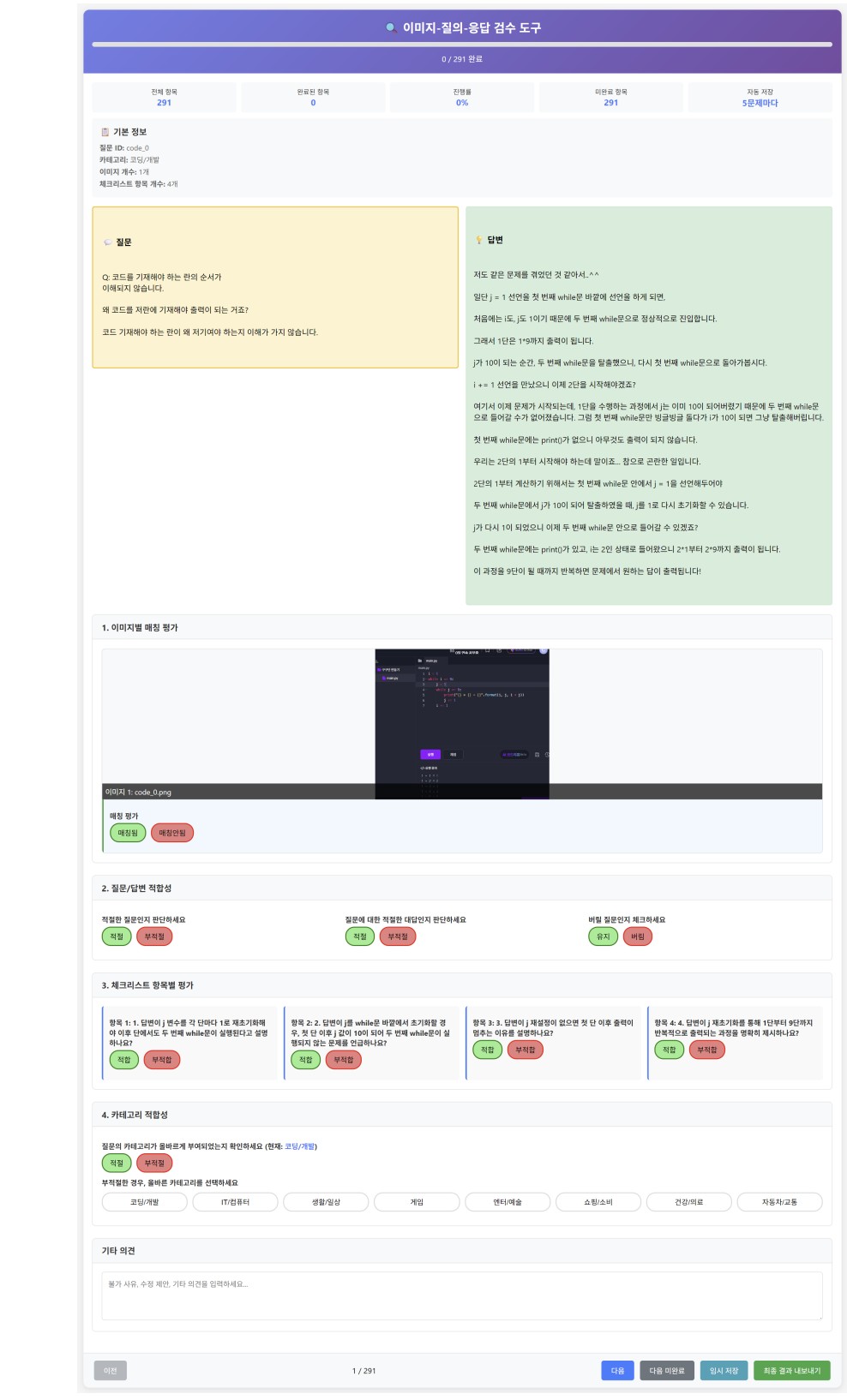

Figure 2: Screenshot of our in-house web annotation tool (Phase 1 of Stage 6). The interface (shown in Korean) allowed annotators to assess image relevance, question/answer appropriateness, checklist accuracy, and category assignment. This ensured rigorous and consistent validation across annotators.