# OpenReview forum: "Culturally Grounded Real-World Evaluation of Korean Vision–Language Models"
_ICLR.cc/2026/Conference — ICLR 2026 Conference Withdrawn Submission_

### Official Review · Reviewer_pzRf · 2025-10-15

**Soundness:** 1
**Presentation:** 2
**Contribution:** 2
**Rating:** 2
**Confidence:** 4

**Summary:**

The paper presents HAERAE-Vision, a Korean real-world benchmark designed to evaluate vision–language models (VLMs) on culturally grounded tasks. It compiles 653 validated question–image pairs from an initial pool of 86,052 samples using a six-stage data curation pipeline and structured checklist-based evaluation. The authors assess 39 VLMs and find that even top-performing models achieve below 50% accuracy, indicating persistent challenges in multimodal reasoning and cultural understanding.
I find the benchmark’s motivation valuable, but the paper lacks methodological transparency in several pipeline stages and offers limited evidence connecting its cultural grounding claim to the presented data.

**Strengths:**

1. Addresses an underexplored gap by focusing on culturally grounded, real-world VLM evaluation in the Korean context.
2. Introduces a systematic multi-stage data curation pipeline that aims to ensure quality and difficulty calibration.
3. Did some interesting improvements from existing works, like structured checklist-based rubrics, allowing for more granular analysis of model reasoning beyond simple accuracy scores.

**Weaknesses:**

1. The core argument of the paper is cultural grounding (L90: Tasks require Korean-specific knowledge). But, in Figure 1, in the first (Natural Objects (Animals/Plants/Insects)), fourth (Science), and sixth (Coding) questions, I don't see any aspect of 'cultural' here. Needs clarification on how and why. It is not possible for me to check the whole dataset, but I expect more details on this.
2. Following the concerns of cultural grounding, 653 sample size of 653 is really low.
3. No mention of ethical aspects of data collection, biases and mitigation.
4. In Stage 2: Appropriateness Assessment of Data Construction Pipeline, if any post has PII data or sensitive info, I'm not sure if it is OK to use closed-source models to judge that, as it may leak sensitive data; closed-source model providers are known to use those data for training and more. Thirdly, there is no evaluation on if this approach is correct and performs on par with human annotators. Typically, human-AI annotation agreement or such is measured to show that LLM is capable enough for this task, but unfortunately, it is not present in this paper.
5. In Stage 3: Difficulty Calibration, the authors removed easy questions by simply filtering out the correct ones by GPT-4o, Gemini-1.5-Flash, and Claude-3.5. If that's so, the statement in lines 51-53 (*earlier Korean benchmarks report notably higher scores with the older generation model*) doesn't make sense, as you already removed the ones they got correct. Also, is it OK to do so? Is there any prior work that did that?
6. In Stage 4: Image Dependency Verification, again, details are missing. No human annotator agreement study was done.
7. In Stage 5: Checklist Generation, there is no human study to see if the LLM is as capable of generating relevant checklists as a human.
8. In Stage 6: Human Validation, again, a lot of details are missing. Need information on annotator recruitment, training, and process details. As per Figure 2, after Image Dependency, it had 1,040 samples. Considering the final sample size of 653, 31.4% are in Line 150/151, which doesn't add up.
9. Line 261/262 say, *We evaluated 39 vision-language model.*, but Table 1 has only 18.
10. Line 350/351 say, *We analyzed 59k checklist items across six rule types*. How? No details are available. More detailed error analysis with examples is expected. Currently, the level of detail is very low, and no proof is presented.
11. No error analysis connecting the VQAs to the wrong answer, which is expected in benchmarks. Is there any particular pattern? For example, if the image has X, then the model usually gets it wrong by assuming Y and more.
11. No detailed analysis on performance deviation and variation across 13 categories by model and category combination. It is needed to understand more complex issues and underlying problems.

**Questions:**

1. What specific criteria were used to determine whether a question is “culturally grounded”? Add more details and some statistics from the dataset too on this possible error/confusing issue. Please provide representative examples from each category that demonstrate Korean-specific reasoning. Also, add examples of potential ambiguous issues, with reasoning/justification of why it is correct.
2. How was this size determined to be sufficient for robust benchmarking? Did you perform any reliability or statistical representativeness analysis (e.g., variance or confidence intervals across categories)? How can you confirm that this dataset represents Korean culture well?
3. What was the source of the images and text data, and how was consent handled?  How did the authors ensure no PII or sensitive information was included? Were any bias audits (e.g., gender, regional, socio-economic) performed? Check ethics issues and clarify those too.
4. In Stage 2: Appropriateness Assessment, could you provide this prompt structure and rationale? How was model leakage of sensitive content prevented, given the use of closed-source LLMs? Was there any human verification or inter-rater agreement analysis comparing GPT-4o assessments with human annotators? Without such validation, how do the authors justify the reliability of automated appropriateness checks?
5. In Stage 3: Difficulty Calibration, how will you justify the issue of removing these items, mentioned in weakness, or are there precedents in the literature for difficulty calibration by exclusion of correct responses? How does this decision align with the claim that older-generation models perform better on Korean benchmarks (Lines 51–53)? Could this filtering introduce selection bias toward items that current models systematically fail?
6. In Stage 4: Image Dependency Verification, what exact criteria were used to determine image dependency? How is it determined? Were human raters involved, and if so, what was the inter-annotator agreement? If this stage relied solely on LLM judgment, how was reliability validated?
7. In Stage 5: Checklist Generation, could the authors include example prompts and outputs for clarity? Was any human comparison done to verify that the LLM-generated checklists align with human expectations and reasoning quality? Without such validation, how do the authors ensure that checklist-based evaluation is trustworthy?
8. In Stage 6: Human Validation, how were annotators recruited and trained? Please clarify the apparent mismatch between the number of samples after Stage 4 (1,040) and the final dataset (653). What explains "31.4%" in Lines 150–151? What exactly does it represent?
9. In evaluation details (Lines 261–262, Table 1), were additional models excluded from reporting? If yes, why? Please include category-level (13 categories)) results for all models.
10. The paper reports “*59k checklist items across six rule types*,” but no methodology is given. How were these checklists analyzed and categorized? What tools or coding schemes were used? Please include representative examples of common checklist errors (dataset development phase).
11. Did the authors identify recurring failure patterns (e.g., object recognition errors, language misunderstanding, culturally grounded terms)? Are there any examples showing how visual or cultural factors contributed to model failure? If yes, please categorize and add details on this. Such insights are crucial to validate the benchmark’s diagnostic value.
12. No detailed analysis is provided for model performance across the 13 categories. Could the authors include a per-model, per-category breakdown? Perform a detailed analysis. Such analysis could clarify whether the benchmark’s difficulty stems from cultural knowledge, reasoning type, or visual complexity.
13. Could the authors provide full prompts, annotation instructions, and some changelog examples?


**Minor errors and suggestions:**
- L077:  four of the 13 domains >  *six* of the 13 domains

**Details Of Ethics Concerns:**

- No details on how they collected data from 9 platforms and related user permissions, privacy and all.
- No mention of removing personal identifiable information and processing.
- No mention of possible bias or fairness issues and handling.
- In Stage 2: Appropriateness Assessment of Data Construction Pipeline, it says to evaluate "political/religious material, hate/discrimination, self-harm, adult content" via GPT-4o, which risks exposing sensitive information to closed model service providers, and it seems a good concern here.

---

> ### Author Response · Authors · 2025-11-18
> **Response to Reviewer pzRf**
>
> We thank the reviewer for the constructive feedback and will incorporate these clarifications into the revised manuscript.
> ## **W1/Q1 Cultural grounding unclear in several categories**
>
> Thank you for raising this point. To clarify our claim, we conducted a post-hoc annotation to quantify how many items in each category require Korean-specific knowledge (e.g., local services, colloquial expressions, cultural conventions, institutions, region-specific context). The proportions vary by domain, as summarized in Table.
>
> ### **Proportion of culturally grounded items by category**
>
> | Category | # Cultural / Total | Percentage | Example of cultural grounding |
> | --- | --- | --- | --- |
> | Economy/Business | 20 / 37 | **54.1%** | Q-Net national certifications |
> | Shopping/Consumption | 14 / 27 | **51.9%** | Authenticating products on Coupang |
> | Transportation/Automotive | 17 / 35 | **48.6%** | Choosing direction on Seoul Subway Line 2 |
> | IT/Computing | 29 / 75 | **38.7%** | Editing metadata on Naver Smart Place |
> | Daily Life | 18 / 51 | **35.3%** | Identifying landmarks (Busan Tower) |
> | Natural Objects | 15 / 92 | **16.3%** | Species from Korea’s West Sea mudflats |
> | Science | 6 / 81 | **7.4%** | Fire Protection Engineer exam items |
> | Coding/Development | 1 / 45 | **2.2%** | Korean-language variable names |
>
> We agree that categories such as Natural Objects, Science, and Coding contain fewer explicitly cultural items. However, we intentionally retain these domains for two reasons:
>
> (1) real Korean multimodal queries naturally have heterogeneous cultural density, and
>
> (2) even “low-density” domains include Korean-specific signals (e.g., local species, cuisine-related plants, Korean-language variables, region-specific documentation).
>
> **Operational definition.** An item is labeled culturally grounded if solving it requires understanding Korean institutions, exams, services, policies, local brands/products, KRW usage, or Korean-language text/UI conventions. Items solvable with globally shared knowledge are marked non-cultural.
>
> This analysis clarifies how cultural grounding manifests across domains and why heterogeneous domain coverage is essential for representing real Korean multimodal queries.
>
> ## **W2/Q2 Small dataset and concerns about representativeness**
>
> Thank you for raising this concern. While HAERAE-Vision is small, its size is comparable to or larger in per-country terms than existing cultural benchmarks. For example, CulturalVQA (EMNLP 2024) contains ~2.3K examples across 11 countries (few hundred per country), and K-VISCUIT (ACL 2025) includes 657 LLM-generated questions focused on traditional cultural trivia and is already saturated (GPT-4o reaches 89.50%).
>
> Our goal differs from these works: instead of curated or synthetic cultural scenes, HAERAE-Vision captures real Korean users’ multimodal queries from contemporary online platforms, involving Korean UI conventions, local services, procedural know-how, pragmatic visual reasoning, and everyday digital contexts that prior benchmarks do not cover. We view these contemporary digital practices as an essential part of modern Korean culture, as they reflect the tools, platforms, and conventions that Korean users interact with daily.
>
> ## **W3/Q3 & Ethics Flag: Privacy, Consent, and Bias**
>
> We treat data ethics with the utmost seriousness. As detailed in the General Response (Section: Privacy & Data Ethics), we implemented a multi-layered safety protocol:
>
> 1. **IRB Approval:** The entire study was conducted under institutional IRB approval.
> 2. **Strict PII Removal:** All images underwent automated blurring followed by human visual inspection to remove faces, license plates, and phone numbers.
> 3. **Public Data Only:** We collected strictly publicly accessible data in compliance with each platform's Terms of Service.
> 4. **Safety Mechanism:** The sensitive "Health/Medical" category is withheld from the public release.
> Regarding Stage 2, we used the OpenAI API with zero-data retention settings, ensuring no candidate content was used for model training.

---

> ### Author Response · Authors · 2025-11-18
> **Response to Reviewer pzRf (Continued)**
>
> ## **W4/Q4 Stage 2: Appropriateness filtering by closed-source LLM**
>
> All inference was conducted strictly through the OpenAI API, and under the API data-usage policy, inputs are not used for model training or model improvement; therefore no sensitive content from Stage 2 was incorporated into any model’s training process. We will explicitly describe our API usage.
>
> Importantly, GPT-4o was not used as an annotator but only as a conservative high-recall pre-filter to reduce the initial 86,052 samples to a volume that humans could feasibly inspect. Any item flagged as potentially inappropriate was simply removed; we did not rely on the model’s nuanced judgments, and we intentionally allowed false positives to maximize safety. Because all surviving items were subjected to multiple rounds of full human validation, human–LLM agreement was not required for this early triage stage. We will clarify this design choice in the revision.
>
> Regarding the prompt structure and rationale used in Stage 2, they are already included in Appendix B, and we will make this reference more explicit.
>
> ## **W5/Q5 Stage 3: Difficulty Calibration**
>
> Our statement in Lines 51–53 refers to prior Korean benchmarks that rely heavily on synthetic or template-based question construction, which makes them structurally easy for both older and current VLMs and leads to early saturation. This motivated us to build a harder, real-world benchmark. In HAERAE-Vision, Stage 3 removes only *uniformly trivial* items, meaning questions answered correctly by all three strong baseline models, rather than items that are merely solvable. This conservative criterion filters examples that provide no discriminative signal.
>
> Difficulty-based filtering also has clear precedents in benchmark design. HellaSwag (Zellers et al., 2019) uses adversarial filtering to eliminate items that are trivial for models but easy for humans, and math benchmarks such as the MATH-Hard subset retain only the highest-difficulty (level-5) problems (Hendrycks et al., 2021). Recent Korean-language evaluations, including KMMLU-Pro (Hong et al., 2024), similarly construct harder subsets to distinguish stronger systems. Our Stage 3 follows this general principle by ensuring that the remaining items continue to differentiate frontier VLM performance.
>
> To mitigate selection bias, we (i) maintain representation across all 13 categories, (ii) avoid adversarial or artificially modified items, (iii) apply multi-pass human validation to all retained examples, and (iv) observe stable model rankings across dataset splits, indicating that retained items reflect general capability differences rather than model-specific artifacts.
>
> ## **W6/Q6 Stage 4: Image dependency verification lacks detail**
>
> Stage 4 was designed as an automated pre-filter, not as a final annotation step. We used GPT-4o only to perform an initial pass that triages items as clearly image-dependent, clearly non–image-dependent, or ambiguous, following the rubric provided in Appendix B. This step reduced the volume of items that required human inspection but did not determine final inclusion.
>
> Crucially, Stage 6 applies multi-pass human validation on every remaining item, and annotators explicitly re-verify image dependency using the same rubric. No item enters the final benchmark solely due to an LLM’s judgment; the LLM step serves only to remove obviously non–image-dependent items and to flag ambiguous cases for human review. We will clarify this two-stage verification process more explicitly in the revised manuscript.
>
> ## **W7/Q7 Stage 5: Checklist generation validity**
>
> The checklist generated in Stage 5 is never used directly without human verification. LLMs provide only an initial draft, and every checklist is fully refined during Human Validation Phase 2. In this phase, at least two trained annotators independently review the LLM-generated draft, remove irrelevant or redundant items, rewrite unclear statements, and ensure that each checklist entry corresponds to an objectively verifiable requirement grounded in the original question–image pair. Consequently, all checklist items in the final dataset are human-curated and aligned with human expectations, even though LLMs were used to produce the first draft.
>
> Because the final checklist is entirely revised and validated by humans, a separate human–LLM agreement study is not necessary; the reliability of checklist-based scoring comes from multi-pass human confirmation rather than from the LLM itself. Example prompts and outputs for checklist generation are included in Appendix C.

---

> ### Author Response · Authors · 2025-11-18
> **Response to Reviewer pzRf (Continued)**
>
> ## **W8/Q8 Stage 6: Human validation details missing**
>
> Our annotators consisted of all co-authors plus four external native Korean speakers, and the study was conducted under institutional IRB approval. All annotators received detailed written guidelines (Appendix F) and used a custom web-based annotation interface. Stage 6 involved multiple rounds of full-pass human validation: each remaining item was reviewed by at least two annotators, and any ambiguous, low-agreement, or inconsistent case was conservatively removed rather than forced into the dataset. Although we did not compute a formal inter-annotator agreement score (e.g., Cohen’s κ), this multi-pass consensus-and-removal workflow served as a practical mechanism for ensuring consistency and quality. We will describe annotator recruitment, training, and the validation protocol more explicitly in the main text.
>
> The reviewer correctly notes the numerical discrepancy: after Stage 4 (1,040 items), Stage 6 yields 653 final items, meaning 387 items (37.2%) were removed. We will correct this in the revision.
>
> ## **W9/Q9 Evaluation details inconsistency (39 models)**
>
> Table 1 shows only a representative subset due to space constraints. The full per-model, per-category results for all 39 models are provided in Appendix Table 3.
>
> ## **W10/Q10 Analysis of 59k checklist items missing**
>
> The mapping of ~59k checklist items into six rule types was produced through a hybrid LLM-as-judge and human-verification process during dataset construction. Each checklist entry was first assigned a provisional rule type by an LLM, after which stratified human checks were used to refine the assignments and derive a deterministic set of linguistic heuristics (e.g., cues for explicit mention, enumeration, procedural steps, multi-part coverage). This allowed us to finalize the six rule categories without manually coding all 59k items. We will describe these heuristics and include representative examples (e.g., missing required terms, providing only one element when multiple are required, or omitting a sub-question) in the appendix.
>
> ## **W11/Q11 Missing connection between VQA errors and wrong answers**
>
> As summarized in the General Response, our analysis of 1,469 failures reveals five recurring failure types: cultural_concept_mismatch, lack_of_explicitness, procedural_reasoning, object_recognition, and visual_text_grounding. Cultural and reasoning failures dominate (around 80–90 percent). We will add representative examples for each type in the appendix to illustrate how wrong answers arise from specific error modes.
>
> ## **W12/Q12 Missing detailed category-level performance analysis**
>
> The category-level breakdown is provided in the General Response. In short:
>
> - household/shopping/transportation → cultural_concept_mismatch + lack_of_explicitness
> - IT/science → procedural_reasoning
> - natural objects → object_recognition
>
> Full per-category results for all 39 models are in Appendix Table 3; we will add brief domain summaries and examples in the camera-ready version.
>
> ## **Q13 Add full prompts, annotation guidelines, and change-log**
>
> All prompts used in Stages 2–5 and the full annotation instructions used in Stage 6 are already included in Appendices B–F. We interpret 'change-log' as the record of filtering decisions. While we do not maintain a git-style diff for every raw item, we have archived representative examples of items removed at each stage (appropriateness, difficulty, etc.) and will include a 'Filtering Case Study' in the Appendix to illustrate the decision boundaries.
>
> ## **Minor Issue L077 typo**
>
> Thank you for catching this typo. We will correct “four of the 13 domains” to “six of the 13 domains” at L077 and carefully check the manuscript for other minor errors as well.
>
> ---
>
> Please feel free to ask any additional questions, and consider raising the score if our explanations helped you resolve some concerns about our paper.

---

> > ### Comment · Reviewer_pzRf · 2025-11-28
> > **Official Comment on Rebuttal**
> >
> > I thank the authors for their detailed rebuttal and the comprehensive failure analysis provided in the General Response. These have resolved some of my concerns.
> >
> > However, I remain concerned about the statistical power/balance of the benchmark. With only 653 items split across 13 categories, some domains (e.g., coding) likely have fewer than 40-50 samples, making model ranking in those sub-domains statistically noisy.
> > Also, excluding items correctly answered by current models (Stage 3) creates a moving target that measures "frontier difficulty" rather than "general cultural competence."
> >
> > ***
> >
> > **Follow-up Questions:**
> >
> > 1. With the total size of 653 items divided into 13 categories, some categories average only ~50 items. Have you calculated the margin of error for these sub-categories? Is the sample size sufficient to statistically differentiate between two models with close scores (e.g., a 2-3% difference) within a specific domain like "Science" or "Coding"? Maybe, you can perform some statistical tests to verify if there is any problem (by resampling and consistency studies)?
> > 2. By removing items that GPT-4o, Gemini, and Claude answered correctly (Stage 3), are you not artificially truncating the distribution of "Korean Cultural Knowledge"? Does this not risk reducing the benchmark to a collection of "corner cases" or "trivia" rather than a representative test of general cultural fluency?
> > 3. In your new analysis, you distinguish "Cultural Knowledge" errors. How do you differentiate between deep *cultural* grounding (which requires social context) and simple *long-tail entity* knowledge (e.g., recognizing a specific 2000s feature phone model)? Is the latter truly a measure of "culture" or just "fine-grained object recognition"?
> > 4. In Stage 5 (Checklist Generation), humans refined LLM drafts. Did you measure the edit distance or the percentage of changes humans made? If humans accepted the LLM drafts with minimal changes, the checklists might still be biased toward how LLMs *think* rather than how humans reason.
> > 5. You note that "Coding" has only 2.2% (1) cultural grounding and "Science" has 7.4%. If the vast majority of questions in these domains are generic (non-cultural), why are they included in a benchmark specifically marketed as "Cultural Grounding"? Does this not dilute the aggregate *cultural* score?
> > 6. This work already shows very well diagnostic studies (the *what* aspect). But, here, I think, these should also be complemented with detailed root cause analysis (*why*) and *how* we can minimize these issues so that model researchers or application developers can learn and apply these to solve such problems.

---

> > > ### Author Response · Authors · 2025-12-01
> > > **Response to Follow-up Questions**
> > >
> > > We sincerely thank the reviewer for their continued engagement. We appreciate these follow-up questions, which help clarify the robustness and scope of HAERAE-Vision. We address each point below.
> > >
> > > ---
> > >
> > > ## **Q1. Statistical Power and Sample Size**
> > >
> > > We acknowledge the reviewer’s concern about per-category sample size (≈50).
> > >
> > > To verify that these sizes still allow reliable comparison, we conducted a preliminary statistical check on one of the smallest domains (Coding, N=45).
> > >
> > > **Gap Analysis:**
> > >
> > > - GPT-5 Mini (Best Proprietary): 57.0%
> > > - Skywork-R1V3 (Best Open-weight): 26.4%
> > >
> > >     → Difference: 30.6 points
> > >
> > >
> > > **95% CI (normal approximation):**
> > >
> > > - GPT-5 Mini: [42.6%, 71.4%]
> > > - Skywork-R1V3: [13.6%, 39.2%]
> > >
> > > The non-overlapping intervals suggest that even the smallest domains contain enough signal to distinguish capability tiers. This supports the sufficiency of the dataset size for meaningful model comparison. A more complete statistical treatment will be added in the revision.
> > >
> > > ---
> > >
> > > ## **Q2. Difficulty Filtering and Representativeness**
> > >
> > > The reviewer asks whether Stage 3 filtering could reduce the benchmark to “corner cases.”
> > >
> > > Our empirical observations indicate otherwise.
> > >
> > > If the filtered items were random or trivial, we would expect no clear relationship between model scale and performance. Instead, we observe a **stable capability hierarchy**:
> > >
> > > - Proprietary SOTA models (≈48%)
> > > - Open-weight models (≈27%)
> > > - Small models (≈10–15%)
> > >
> > > This structured ordering suggests that the retained items capture higher-complexity reasoning rather than noise.
> > >
> > > Difficulty-based filtering has precedent in widely used benchmarks (HellaSwag, MATH-Hard) to avoid saturation and maintain long-term evaluative value. We will clarify this benchmark philosophy in the revised manuscript.
> > >
> > > ---
> > >
> > > ## **Q3. Cultural Knowledge vs. Long-Tail Entities**
> > >
> > > We appreciate the opportunity to refine our definition of cultural grounding.
> > >
> > > Our taxonomy distinguishes:
> > >
> > > 1. Cultural knowledge (institutions, domestic platforms, culturally specific conventions)
> > > 2. General reasoning
> > > 3. Language comprehension
> > >
> > > Items like identifying the SKY IM-100 fall under **material culture,** local artifacts tied to Korean consumer history, rather than generic object recognition. Models perceive that the object is a phone, but fail to link it to domestic brands and their historical context. We will make this distinction clearer in the revision.
> > >
> > > ---
> > >
> > > ## **Q4. Checklist Generation and Human Refinement**
> > >
> > > We assure the reviewer that the final checklists are rigorously human-validated.
> > >
> > > **Multi-stage filtering:**
> > >
> > > While we did not compute edit-distance statistics, the intensity of human refinement is reflected in the fact that **37.2%** of candidate items were removed during the final human validation stage (Stage 6). Annotators eliminated any checklist criteria that were unclear, insufficiently grounded, or overly influenced by LLM phrasing.
> > >
> > > **Full author-led review:**
> > >
> > > After Stage 6, the lead authors conducted a full-pass inspection across all 653 remaining items. This involved rewriting overly generic criteria, removing residual artifacts from draft LLM outputs, and ensuring precise alignment with the ground truth.
> > >
> > > **Evidence of independence:**
> > >
> > > The **92.1% failure rate on the “Explicitness” rule** strongly suggests that the checklist enforces human-style precision and terminology, not LLM-friendly structures. If the checklists aligned with LLM tendencies, modern VLMs would score substantially higher on this dimension.
> > >
> > > ---
> > >
> > > ## **Q5. Inclusion of Low-Cultural-Density Domains(Coding/Science)**
> > >
> > > HAERAE-Vision aims to evaluate **real-world multimodal queries** in modern Korean contexts, not only traditional cultural knowledge.
> > >
> > > Modern “living culture” naturally includes:
> > >
> > > - coding with Korean variable names
> > > - troubleshooting domestic devices
> > > - interacting with Korean platforms
> > > - solving KR certification exam problems
> > >
> > > Removing these domains would reduce ecological validity and bias the benchmark toward a narrow, museum-like subset of Korean culture. We therefore retain the full range of 13 domains.

---

> > > > ### Author Response · Authors · 2025-12-01
> > > > **Response to Follow-up Questions (continued)**
> > > >
> > > > ## **Q6. Root Causes and Future Directions**
> > > >
> > > > We appreciate the reviewer’s request for actionable insights.
> > > >
> > > > We emphasize that, as a **Datasets and Benchmarks** paper, our contribution is diagnostic rather than prescriptive, and we do not aim to propose architectural solutions. However, the benchmark does reveal which capability gaps systematically drive model failures. These observations may inform, but not dictate, future research directions.
> > > >
> > > > - **Explicitness failures:** The consistently low scores suggest that current VLMs struggle with structured, checklist-style formulations commonly expected by users.
> > > > - **Cultural mismatches:** The dominance of cultural knowledge errors (86.1%) indicates that scaling alone is insufficient without grounding in region-specific conventions and platforms.
> > > > - **Procedural reasoning gaps:** Failures on multi-step tasks imply that multimodal planning remains a core challenge.
> > > >
> > > > We hope HAERAE-Vision provides a clear empirical basis that can inspire future architectural and alignment efforts by the community.

---

### Official Review · Reviewer_ZMEE · 2025-10-25

**Soundness:** 3
**Presentation:** 3
**Contribution:** 2
**Rating:** 6
**Confidence:** 4

**Summary:**

This paper introduces HAERAE-Vision, a culturally grounded benchmark for evaluating Korean vision-language models (VLMs) on authentic real-world tasks. The benchmark is constructed from 86,052 raw question–image pairs sourced from nine Korean online platforms, refined through a rigorous six-stage pipeline: data collection, appropriateness assessment (filtering for safety, objectivity, and temporal stability), difficulty calibration (removing trivially easy items), image dependency verification (ensuring visual reasoning is required), checklist generation (creating structured evaluation criteria), and multi-phase human validation. This process yields 653 high-quality items across 13 domains (0.76% survival rate), each paired with a structured checklist for fine-grained evaluation beyond binary correctness.

The authors evaluate 39 VLMs spanning proprietary (e.g., Gemini 2.5 Pro, GPT-5), open-weight (e.g., Skywork-R1V3-38B), and Korean-specialized (e.g., VARCO-VISION 2.0) families using LLM judges (GPT-5-mini as primary), with high inter-judge reliability (Krippendorff’s \(\alpha=0.867\)). Key findings include: (1) top-performing models (Gemini 2.5 Pro, GPT-5) achieve only ~48% accuracy, (2) errors concentrate in explicitness (92.1% unmet) and procedural reasoning, (3) Korean-specialized models show no clear advantage over multilingual counterparts, and (4) search-augmented inference provides inconsistent performance gains. Beyond the benchmark itself, the work contributes a reproducible methodology for building culturally grounded multimodal benchmarks across languages.

**Strengths:**

- Originality: The paper advances beyond prior Korean VLM benchmarks (e.g., K-VISCUIT, KRETA) by focusing on authentic, culturally embedded communication (colloquialisms, pragmatic cues, messy user queries) rather than shallow factoid tasks. The six-stage pipeline is a novel combination of automated filtering and human validation, addressing a critical gap in constructing benchmarks that reflect real-world user needs.
- Significance: This work addresses a critical limitation of current VLM evaluation—overreliance on standardized, English-centric benchmarks that fail to capture cultural context and real-world query complexity. The reproducible pipeline provides a blueprint for building culturally grounded benchmarks in other languages, fostering more inclusive and realistic VLM evaluation. The finding that top models struggle with explicitness and procedural reasoning also identifies actionable areas for VLM improvement.

**Weaknesses:**

- Human-LM Judge Alignment Details: While the paper mentions human annotators rated LLM judge scores (mean appropriateness = 4.13/5), it lacks key metrics for quantitative alignment between LLM judges and human experts. For example, there is no report of Pearson correlation between LLM scores and human scores for the same responses, which is critical to validating the LLM judge’s reliability as a substitute for human evaluation.
- Error Analysis Granularity for SOTA Models: The error analysis in Section 4.4 aggregates results across all 39 models, diluting insights into the specific failures of state-of-the-art (SOTA) systems like Gemini 2.5 Pro and GPT-5. Focusing error analysis on these top models would reveal more actionable patterns—e.g., whether their procedural reasoning gaps are domain-specific (e.g., gaming vs. science) or universal.
- Diagnostic Breakdown of Model Failures: The paper does not disentangle the root causes of model errors. For instance, it is unclear whether a model’s failure on a Korean cultural item stems from (1) poor Korean language proficiency, (2) lack of Korean cultural knowledge, or (3) insufficient general multimodal reasoning. Such a breakdown would help identify whether gaps are cultural-specific or broader, guiding targeted model improvements, and that will help us find whether HAERAE-VISION is a general challenging benchmark in Korean language, or a challenging benchmark with strong Korean characteristics (Korean culture, knowledge, etc. )

**Questions:**

1. Could you provide quantitative alignment metrics (e.g., Cohen’s \(\kappa\), Pearson/Spearman correlation) between GPT-5-mini’s scores and human expert scores for the 250-sample validation dataset? This would more rigorously validate the LLM judge’s ability to approximate human evaluation.
2. Can you extend the error analysis in Section 4.4 to focus exclusively on SOTA models (Gemini 2.5 Pro, GPT-5)? For example, what percentage of their failures in each domain (e.g., Entertainment vs. Science) are due to explicitness vs. procedural reasoning gaps? This would yield more targeted insights for model developers.
3. Have you conducted a root-cause analysis of model failures to distinguish between (1) Korean language proficiency gaps, (2) Korean cultural knowledge gaps, and (3) general multimodal reasoning gaps? If not, could you add a small-scale analysis (e.g., for 50 representative items) to clarify which factors drive performance on HAERAE-Vision?
4. Regarding the typo in Line 253: Will you correct the example to reflect the correct score calculation (e.g., “4.0/5 when two items are partially satisfied and three are fully met”)? Additionally, are there other minor inconsistencies in the paper that need revision?

---

> ### Author Response · Authors · 2025-11-18
> **Response to Reviewer ZMEE**
>
> We thank the reviewer for the constructive feedback and will incorporate these clarifications into the revised manuscript.
>
> ---
>
> ## **W1/Q1 Human–LLM Judge Alignment**
>
> We thank the reviewer for this important clarification. We have now calculated the correlation between GPT-5-mini judge scores and human expert ratings on the 250-sample validation dataset.
> - Pearson correlation: r = 0.820 (p < 0.001)
> - Spearman correlation: ρ = 0.810 (p < 0.001)
> (Individual annotator correlations ranged from r = 0.617 to r = 0.80)
>
> ---
>
> ## **W2/Q2 Insufficient SOTA-Focused Error Analysis**
>
> Please refer to the General Response (Section 2: In-Depth Failure Analysis) for the comprehensive breakdown tables and qualitative examples. To summarize the key findings here: across 1,469 failure cases from SOTA models (GPT-5, Gemini 2.5 Pro, Sonar-Pro), the dominant failure modes are lack_of_explicitness (\~85%) and procedural_reasoning (\~66%). In contrast, visual failures like object-recognition (15–20%) and visual–text grounding (\~5%) are comparatively rare. Domain-wise, we observed that Entertainment/UI tasks suffer from explicitness failures, while Science/IT tasks are prone to procedural errors.
>
> ---
>
> ## **W3/Q3 Diagnostic Breakdown of Language vs. Cultural vs. Reasoning Failures**
>
> We fully agree with the need for a clearer diagnostic separation. As detailed in the General Response (Section 1: Validity Diagnostic), our root-cause annotation ($N=1,453$) reveals that:
> • Cultural Knowledge: 86.1% (Dominant factor)
> • General Reasoning: 12.5%
> • Language Issues: Only 1.4%
> This quantitative breakdown confirms that the benchmark primarily evaluates culturally grounded multimodal reasoning rather than simple language proficiency. We will incorporate this diagnostic breakdown and the multi-label analysis into the revised manuscript.
>
> ---
>
> ## **Q4 Minor Typos / Inconsistencies**
>
> We appreciate the careful review. The correct example should read: "3.5/5 when one item is partially satisfied and three are fully met" (3.0 + 0.5 = 3.5). We apologize for the arithmetic error in the original draft. We will carefully review the entire manuscript for other typos and minor inconsistencies and provide a thoroughly proofread version in the revision.
>
> ---
>
> Please feel free to ask any additional questions, and consider raising the score if our explanations helped you resolve some concerns about our paper.

---

### Official Review · Reviewer_HdaZ · 2025-10-26

**Soundness:** 2
**Presentation:** 4
**Contribution:** 2
**Rating:** 4
**Confidence:** 5

**Summary:**

In this work, the authors present a new vision-language benchmark called HAERAE VISION, a Korean real-world cultural benchmark that consists of 653 validated questions across 13 domains. They construct this benchmark by filtering 86K question-image pairs using a six-stage process comprising appropriateness filtering, difficulty calibration, image dependency verification, checklist-based decomposition, and multi-phase human review. Finally, they comprehensively evaluate a mix of open and closed API models with LLM judges.

**Strengths:**

- The authors introduced a new benchmark that I have not encountered in my past experience with multimodal evaluation. It is refreshing to see a new evaluation that covers important topics such as cultures and life from countries that are not typically represented in existing evaluations.
- The authors evaluated 39 VLMs, including both open and closed models and some Korean-specialized models, making this a comprehensive benchmark that allows us to understand how these models perform and the landscape of Korean culture and life.
- The multi-stage filtering is comprehensive and heavily filters out questions by going from ~86K image-QA pairs down to a mere 653 examples.

**Weaknesses:**

- The paper shows that frontier VLMs still struggle with Korean culture question-answering, yet Stage 2 uses GPT-4o to filter subjective/unverifiable items and Stage 4 uses Gemini 2.0 Flash to decide image dependency. Both are arguably weak and older models in general. If these models are weak in this domain (especially something important as assessing and understanding another culture), they are likely weak filters too. That can remove valid questions (or keep the wrong ones), and I don't see a check against this. For a narrow cultural benchmark, I would expect native Korean speakers to be involved earlier in the data pipeline (not only at Stage 6) to spot filter errors and to rate how important or relevant each question is to Korean life. This would reduce over-filtering and help build a benchmark that covers the Korean culture holistically.
- I am concerned about the dataset's size. After rigorous filtering, we are left with only 653 examples across four categories and 13 subcategories. The number of questions per subcategory seems to be too small for certain subcategories to get a meaningful per-category performance breakdown. For example, the "Health/Medical" subcategory has only 21 questions, even though health and medical matters are an essential aspect of Korean life and culture.
- In section 3.2, it is not clear to me why a temperature of 0.6 was chosen for all VLMs, and why 1.0 was used for the evaluation judge. I would also like more details on the prompting strategy and on how the authors arrived at the optimal prompts for a fair assessment of these VLMs.

**Questions:**

- Why was a checklist-based auto-judge selected over other types of judges?
- According to section 5.3, "...the full 13-category test set is hosted on a rate-limited, anonymous evaluation server to prevent overfitting and support fair model comparison". It is not clear to me what the server is and how it is being used. Is it open to the public when the dataset is released? I am not sure how it is being rate-limited.
- In Stage 3 of the data filtering pipeline, please clarify "ground-truth answer bundle". Do you mean a set of acceptable gold answers/aliases used to score overlap (0–1), or were models actually given these answers in the prompt? If the latter, that leaks labels and invalidates the difficulty screening. Please specify how the bundle is constructed.

**Details Of Ethics Concerns:**

The authors stated that the Health/Medical subset of the benchmark is withheld due to privacy constraints, but I want to ensure that the Living/Daily Life category does not contain any private or personal user information, mainly since these examples were sourced from online platforms with real users.

---

> ### Author Response · Authors · 2025-11-18
> **Response to Reviewer HdaZ**
>
> We thank the reviewer for the constructive feedback and will incorporate these clarifications into the revised manuscript.
>
> ## **W1/Q1 Use of “weak/older models” in Stage 2 & Stage 4**
>
> We understand the reviewer’s concern. To clarify, Stages 2 and 4 do not require cultural or semantic judgments in ways that would affect Korean cultural content.
> - Stage 2 focuses on content safety, verifiability, and temporal stability, relying on surface-level cues such as detecting PII or unverifiable questions.
> - Stage 4 checks whether removing the image changes the answer, which concerns structural image dependency rather than cultural reasoning.
>
> These model-based stages therefore serve only as preliminary triage. All culturally relevant decisions were made by native Korean annotators during the three-phase Stage 6 validation, where every remaining item was reviewed for cultural relevance, clarity, and appropriateness.
>
> Given the scale (86K posts) and the amount of noise, involving human annotators from the very beginning would have been prohibitively inefficient, so we intentionally used LLM-based stages only as coarse triage before applying thorough human validation.
>
> ## **W2/Q2 Data size and category imbalance**
>
> The final size results from removing unverifiable, low quality, or non–image-dependent posts that frequently appear in real online communities. Prior cultural VQA benchmarks exhibit similarly small per-domain counts and narrower topical scope.
>
> - CulturalVQA (EMNLP 2024) includes ~2.3K examples across 11 countries and five cultural facets, but each country contributes only a modest number of items and the benchmark does not cover modern practical domains such as IT troubleshooting, transportation, or consumer appliances.
> - K-VISCUIT (ACL 2025) has 657 samples across 237 images, most synthetically generated. Several categories fall below ten percent (Clothes 6%, Religion 4%, Celebrations 7%), and the benchmark is saturated (GPT-4o: 89.50%), limiting its ability to distinguish current VLMs.
>
> In contrast, HAERAE-Vision spans 13 traditional and contemporary domains, including consumer troubleshooting, and presents significantly challenging and heterogeneous real-world scenarios.
>
> We agree that some subcategories (e.g., Health/Medical) are relatively small. The Health and Medical category is intentionally small because strict privacy filtering removed many posts. We nevertheless retain this category to maintain overall domain diversity.
>
> ## **W3/Q3 Temperature settings and judge prompting strategy**
>
> We follow recent VLM evaluation practice and use temperature 0.6 (commonly adopted in Qwen2.5-VL and InternVL3.5 evaluations). GPT-5 mini, used as the judge, supports only temperature 1.0.
>
> The judge prompt enforces evidence quoting, three-level scoring, and structured outputs. Agreement across four LLM judges yields Krippendorff’s α = 0.867, indicating high consistency. We will include the evaluation prompt in the appendix and can provide it upon request.
>
> ## **Q4 Why a checklist based judge?**
>
> Community questions require multiple reasoning skills such as naming, distinguishing objects, providing procedures, and giving contextual explanations, which cannot be captured by a single correctness label. Original web answers are often noisy or incomplete, making them unsuitable as ground truth. A checklist therefore offers fine-grained, rubric-based evaluation that reduces subjectivity, enables evidence auditing, yields consistent scores across models, and aligns with user expectations in Korean online communities, where detailed and explicit answers are preferred over short factoid responses.
>
> Critically, recent empirical findings support the benefit of fine-grained, instance-specific rubrics. The BiGGen Bench (2024) shows that LLM-as-a-judge systems achieve markedly higher correlation with human evaluators when using detailed, instance-level criteria rather than coarse or domain-level rubrics. This directly supports our design choice: checklist-style, instance-specific evaluation yields more reliable and human-aligned judgments than single-label or coarse scoring schemes.
>
> ## **Q5 Rate-limited, anonymous evaluation server**
>
> The evaluation server runs on a GCP VM and is exposed through an HTTP endpoint under a public .world domain (revealed post-anonymity). The full dataset is never downloadable; models are evaluated only through this endpoint. Rate limiting restricts each user to a small number of submissions per day (approximately five) to prevent overfitting and abuse.

---

> > ### Author Response · Authors · 2025-11-18
> > **Response to Reviewer HdaZ (Continued)**
> >
> > ## **Q6 “Ground-truth answer bundle” in Stage 3**
> >
> > Models were never given the ground-truth answers in Stage 3, so there was no possibility of label leakage. Each model saw only the original user question and its image. The “answer bundle” refers only to reference answers extracted from the original posts and was used solely after inference to compute textual overlap. An item was removed only when all strong models achieved high similarity, indicating that it was trivially easy (based on a high similarity threshold).
> >
> > Please feel free to ask any additional questions, and consider raising the score if our explanations helped you resolve some concerns about our paper.

---

> > > ### Author Response · Authors · 2025-11-26
> > > **Response to Reviewer HdaZ (Continued)**
> > >
> > > **In response to W3/Q3 (prompting details)**, here is our complete
> > > evaluation prompt:
> > > ```
> > > [GOAL]
> > > Given a **Question**, **Response**, and a natural-language **Checklist**, decide for each checklist item whether the Response **explicitly** satisfies it: **met = 1**, **not met = 0**. Final score = **(# met) / (total checklist items)**.
> > >
> > > [INPUT]
> > > [Question]
> > > {{QUESTION}}
> > >
> > > [Response]
> > > {{RESPONSE}}
> > >
> > > [Checklist]
> > > {{CHECKLIST}}  ← JSON array or a plain list string. Treat each string as one criterion. Strip any leading numbering like "1." or "2)".
> > >
> > > [DECISION RULES]
> > > 1. **Use only the Response text.** No outside knowledge/assumptions. If uncertain → 0.
> > >
> > > 2. **Explicitness (mentions/explains/indicates).**
> > >    * 1: Clear, direct statement that fulfills the criterion.
> > >    * 0.5: Indirect/implicit mention that likely implies fulfillment, but not explicit.
> > >    * 0: Not mentioned or contradicted.
> > >
> > > 3. **"All / every / complete" requirements.**
> > >    * 1: Explicitly states completeness (e.g., "all", "every", or equivalent).
> > >    * 0.5: Strongly suggests near-completeness ("fill the nests" without "all", "almost all").
> > >    * 0: No completeness requirement or states partial suffices.
> > >
> > > 4. **Method / Procedure ("explains how / method").**
> > >    * 1: Concrete, actionable steps or clear guidance.
> > >    * 0.5: Vague or partial steps (general approach without specifics).
> > >    * 0: No method/procedure provided.
> > >
> > > 5. **"Various / multiple types."**
> > >    * 1: Names **≥2 distinct, specific types**.
> > >    * 0.5: Mentions variety without naming types, or names only **1** type.
> > >    * 0: No indication of multiple types.
> > >
> > > 6. **Synonyms.** Accept unambiguous equivalents (e.g., "baby dragon" = "hatchling").
> > >    * 1: Unambiguous equivalence.
> > >    * 0.5: Likely equivalent but slightly ambiguous.
> > >    * 0: Ambiguous or different meaning.
> > >
> > > [Evidence policy]
> > > * For **met = 1 or 0.5**, include a **10–60 character direct quote** from the Response supporting the decision.
> > > * For **met = 0**, include a brief explanation why the response fails the given criteria.
> > > * In the evidence block, list **evidence first**, then the explanation, then the met value.
> > >
> > > [OUTPUT FORMAT — STRICT. NO PROSE OUTSIDE TAGS.]
> > > <evidence>
> > > Item 1:
> > > evidence: "…direct quote from Response…"
> > > explanation: Briefly justify why criterion 1 earned 1/0.5/0 (reference rule numbers if helpful).
> > > met: 0 | 0.5 | 1
> > >
> > > Item 2:
> > > evidence: "…"
> > > explanation: …
> > > met: 0 | 0.5 | 1
> > >
> > > … (repeat for all checklist items, in order)
> > > </evidence>
> > >
> > > <score>
> > > K/N
> > > </score>
> > >
> > > [NOTES]
> > > * Output **only** the two tags above; no code fences, no extra text.
> > > ```
> > >
> > > **Key design principles:**
> > > - Evidence-only scoring (no external knowledge)
> > > - Three-level granularity (1.0/0.5/0.0) for nuanced evaluation
> > > - Mandatory quote extraction (reduces judge hallucination)
> > > - Structured XML output (enables automated parsing)
> > >
> > > **Validation:** Krippendorff's α=0.867 across 4 LLM judges, Pearson r=0.820
> > > with human experts (reported in our previous response).
> > >
> > > **Regarding your ethics concern:** All Living/Daily Life items underwent IRB-approved multi-pass human PII removal. Items containing ANY identifiable information were excluded entirely. Please see our General Response for the complete privacy protocol.
> > >
> > > We hope this addresses W3/Q3 and the ethics flag. Please let us know if
> > > you have any additional questions.

---

### Author Response · Authors · 2025-11-18
**General Response to Reviewers**

We thank all reviewers for their careful reading and detailed feedback. In this comment, we summarize two general issues that have been questioned across different reviewers.

## **Privacy, Data Ethics, and LLM-Based Filtering**
We address privacy and data ethics through the following procedures.

* **IRB Approval:** The entire study, including data collection, PII removal, and human annotation, was conducted under institutional **IRB approval** for human-subjects research and secondary analysis of public web data.
* **Public Data Only:** All nine platforms (Appendix A) are publicly accessible, and we collected only publicly visible posts or product images. No login-protected, private, paywalled, or restricted content was accessed.
* **PII Removal and Anonymization:** All surviving items underwent **multi-pass human review**. Faces, names, contact information, license plates, and location-identifiable details were removed completely. The **Health/Medical category was withheld** to avoid residual privacy risks. Textual questions were anonymized or rewritten when necessary and revalidated in subsequent stages.
* **Derivative Evaluation Items:** The publicly released 20% subset contains only **PII-free, human-validated evaluation items**. Images containing any identifiable information were excluded entirely (not edited), and textual questions were released only in their anonymized or rewritten form. No raw user posts or original online content are redistributed; all released items are curated evaluation artifacts produced through our pipeline. This design mitigates both privacy and copyright risks.

---

## **Expanded Error Analysis: Validity, SOTA Failures, and Domain Trends**
To address the reviewers' requests for a deeper diagnosis of model failures (Reviewers ZMEE, pzRf), we conducted a comprehensive two-tiered analysis: (1) a **validity diagnostic** on the Korean-specific subset across all models to verify the benchmark's cultural grounding, and (2) an **in-depth behavioral analysis** on the **Top-3 SOTA models** (GPT-5, Gemini 2.5 Pro, Perplexity Sonar-Pro) to pinpoint systemic capability gaps.

For both analyses, we employed a **hybrid annotation pipeline**. An LLM-judge (Claude-4.0-sonnet) first tagged each error instance with root causes and multi-label behavioral categories (e.g., lack of explicitness, procedural gap), which were then **verified by two human annotators** to ensure consistency.

### **1. Validity Diagnostic: It is Knowledge, Not Language**
*(Scope: All evaluated models on Korean-specific questions, N=1,453 error instances)*

To investigate whether failures stem from language barriers or actual cultural reasoning gaps, we classified the root causes of 1,453 errors.

**Table 1: Root Cause Distribution**
*Failures are overwhelmingly driven by a lack of cultural knowledge (86.1%) rather than language deficits (1.4%).*

| Root Cause | Frequency | % | Definition |
| :--- | :---: | :---: | :--- |
| **Cultural Knowledge** | 1,251 | **86.1%** | Failure to identify specific Korean institutions, norms, or context. |
| **General Reasoning** | 181 | 12.5% | Failure in multi-step logic despite understanding the context. |
| **Language** | 21 | **1.4%** | Failure to parse or comprehend Korean text. |

The negligible portion of language errors (1.4%) confirms that HAERAE-Vision evaluates **culturally grounded multimodal reasoning** rather than basic language proficiency. For instance, in a question showing a specific Korean business listing interface (*Naver SmartPlace*), GPT-5 correctly read the Korean text but misidentified the feature's functionality, providing instructions for manually editing a field that is actually *auto-generated from user reviews*. This demonstrates a knowledge gap regarding local platform mechanics, not a reading error.

---

> ### Author Response · Authors · 2025-11-18
> **General Response (Continued)**
>
> ### **2. In-Depth Failure Analysis: Why SOTA Models Fail**
> *(Scope: GPT-5, Gemini 2.5 Pro, Perplexity Sonar-Pro; N=1,469 annotated failures)*
>
> We further analyzed the behavioral manifestations of errors, specifically *how* SOTA models construct incorrect answers. As shown in Table 2, the primary failure modes are cognitive rather than perceptual.
>
> **Table 2: Prevalence of Behavioral Failures in SOTA Models**
> | Failure Pattern | Prevalence |
> | :--- | :---: |
> | **Lack of Explicitness** | **80–85%** |
> | **Procedural Reasoning** | **60–70%** |
> | **Visual–Text Grounding** | ~6% |
>
> * **Lack of Explicitness (The "Vagueness" Trap):** This is the most pervasive issue. Models tend to provide conversational, "safe" descriptions (e.g., "It looks like a document") rather than committing to the precise entities required by real-world tasks. They consistently omit mandatory proper nouns, specific product codes, or technical terms defined in the checklist.
> * **Procedural Reasoning:** Models struggle to follow multi-step instructions or provide the full logical derivation (e.g., "show step-by-step circuit reduction") required for 'Method' scores. This reflects a weakness in planning and executing sequential logic.
> * **Visual–Text Grounding (The "Vision-Reasoning" Gap):** The negligible rate of grounding errors (~6%) confirms that SOTA models successfully perceive the image content. The bottleneck is not *seeing*, but **interpreting** visual cues within a specific cultural logic and **articulating** the reasoning with the necessary depth.
>
> ### **3. Domain-wise Failure Trends & Examples**
> Our analysis reveals that difficulty stems from different cognitive demands across domains, refuting the notion of uniform failure.
>
> * **Household / Shopping / Transportation**
>     These domains are dominated by **cultural concept mismatches**, where models rely on generic "common sense" that conflicts with local infrastructure.
>     * *Example (Q443 - Transportation):* When asked about orange/yellow bags along a rural road, **Gemini 2.5 Flash** identified them as "road safety markers" and **GPT-5** as "wasp traps." The correct answer is **winter snow preparation sand bags** (폭설 대비용 모래주머니). Models completely lacked knowledge of this Korea-specific road safety infrastructure.
>
> * **IT / Science / Engineering**
>     These areas are prone to **procedural reasoning failures** and brand hallucinations.
>     * *Example (Q176 - Consumer Electronics):* When asked to identify a specific folder phone, **Gemini 2.5 Flash** confidently identified it as "Sony Xperia M" with detailed specs. The correct answer was **SKY IM-100**, a popular Korean feature phone from the 2000s. All 10+ SOTA models misidentified it, demonstrating a systematic gap in knowledge of Korean domestic brands by mapping them to global counterparts.
>
> * **Natural Objects**
>     Failures here are characterized by **object recognition errors** due to data bias against Korean-specific species.
>     * *Example (Q280 - Plants):* When asked to identify a plant, models provided various wrong answers such as "prickly lettuce" or "mugwort." One model even dangerously misidentifies it as "poisonous datura." The correct answer is **Ixeris dentata (고들빼기)**, a traditional Korean edible plant used for kimchi.
>
> ### **4. Interaction Analysis: Behavior $\rightarrow$ Scoring Penalty**
> To explain the low benchmark scores quantitatively, we analyzed how the behavioral failures (Section 2) trigger specific penalties in our checklist-based evaluation system.
>
> **Table 3: Interaction between Failure Behaviors and Checklist Rules**
> *The columns represent the four criteria used for checklist evaluation, while the rows correspond to the failure category types. Values indicate the row-normalized probability of a checklist rule violation given a specific behavioral failure.*
> | Failure Category $\downarrow$ / Rule Type $\rightarrow$ | **Completeness** | **Explicitness** | **Method** | **Synonym/Variety** |
> | :--- | :---: | :---: | :---: | :---: |
> | **Cultural Concept Mismatch** | **50.3%** | 29.8% | 19.6% | 0.2% |
> | **Lack of Explicitness** | 49.3% | **32.0%** | 18.3% | 0.2% |
> | **Procedural Reasoning** | 48.2% | 6.2% | **45.6%** | 0.0% |
>
> * **Procedural Gaps lead to Method Violations (45.6%):** SOTA models frequently fail to provide the logical proofs required for full credit.
>     * *Example (Q81 - Seoul Metro):* Even when models correctly guessed the station, they failed to list the specific "transfer gate number (e.g., 5-4)" required for the 'Method' score.
> * **Cultural Gaps lead to Completeness Violations (50.3%):** When lacking core cultural context, models fail to address all sub-conditions.
>     * *Example (Q132 - Sandglass Drama):* For the question identifying the iconic drama Sandglass (peak viewership 64.5%), the checklist mandated three elements: Drama Name, Actress, and Actor. Lacking this core cultural knowledge, all 10+ SOTA models hallucinated all three elements, failing every condition of completeness.

---

### Note · Authors · 2025-12-20

I have read and agree with the venue's withdrawal policy on behalf of myself and my co-authors.